

# Cloud identification and classification from high spectral resolution data in the far and mid infrared

Tiziano Maestri[1], William Cossich[1], and Iacopo Sbrolli[1]

[1]Alma Mater Studiorum - Università di Bologna

**Correspondence:** tiziano.maestri@unibo.it

**Abstract.** A new Cloud Identification and Classification algorithm, named CIC, is presented. CIC is a machine-learning algorithm, based on Principal Component Analysis, able to perform a cloud detection and scene classification using a univariate distribution and a threshold, which serves as a binary classifier. CIC is tested on a widespread synthetic dataset of high spectral resolution radiances in the far and mid infrared part of the spectrum simulating measurements from the ESA Earth Explorer

Fast Track 9 competing mission FORUM (Far Infrared Outgoing Radiation Understanding and Monitoring) that is currently (2018/19) undergoing the industrial and scientific Phase-A studies. Simulated spectra are representatives of many diverse climatic areas, ranging from the tropical to polar regions. Application of the algorithm to the synthetic dataset provides high scores for clear/cloud identification, especially when optimisation processes are performed. One of the main results consists in pointing out the high information content of spectral radiance in the far-infrared region of the electromagnetic spectrum to

identify cloudy scenes specifically thin cirrus clouds.

## 1 Introduction

At the end of 2017 the European Space Agency has selected FORUM (Far-infrared Outgoing Radiation Understanding and Monitoring) mission as one of the two instrument concepts to compete for the Earth Explorer 9 satellite program. FORUM is based on a Far-Infrared Spectrometer devoted to high spectral resolution (nominally $0.3\ \mathrm{cm}^{-1}$) measurements from 100 to

1600 $\mathrm{cm}^{-1}$ thus including the so called Far InfraRed (FIR) region, spanning from 100 to 667 $\mathrm{cm}^{-1}$.

The FIR represents an important fraction of Earth's outgoing longwave radiation, which contributes considerably to the planetary energy balance. The atmospheric emission in the FIR is driven by the rotational absorption band of the water vapour molecules and is characterised by strong absorption lines interspersed by narrow regions (called *dirty micro-windows*), where absorption is less intense. The strong absorption features of water vapor roto-vibrational lines cause atmospheric weighting

functions in the FIR to peak in the Middle/Upper Troposphere thus making the on-line upward emission particularly sensible to the atmospheric thermodynamic profile and water vapor content of the highest tropospheric levels. Micro-window radiance is highly sensitive to water vapour mixing ratio (Maestri et al. (2014)) and also affected by the *water vapour continuum absorption*, that is usually modelled through observations (Mlawer et al. (2012), Serio et al. (2000)). Moreover, the condensed phases of water, in form of liquid water and ice clouds, also affect Earth's radiation budget significantly (Sinha and Harries

(1995)) and, in particular, the presence of ice clouds causes lower emitting temperatures and, hence, a shift towards longer



wavelengths (thus towards the FIR) of the peak of the black-body emission distribution function. For this reason, a detailed study of the Earth's radiation budget should account for global, accurate, all-sky conditions measurements of the exiting radiance, including the FIR.

As many recent studies have shown, due to the large sensitivity of the upward radiance in the far infrared part of the spectrum to water vapor and clouds, the FIR can be used to complement standard remote sensing measurements performed in the Mid InfraRed (MIR) part of the spectrum in order to retrieve atmospheric water vapour profile, for cloud detection, classification, and properties derivation. In Merrelli (2012) it is demonstrated that the retrieval of cloud properties and water vapour mixing ratio in the Mid and High Troposphere is more accurate if FIR spectral information (i.e radiance) is considered. Moreover, in Palchetti et al. (2016) it is shown that using the full infrared emission spectrum more information may be retrieved about cirrus cloud microphysical properties. In Maestri et al. (2019) it is shown that far infrared REFIR-PAD channels hold additional information for cloud detection and cloud phase classification from ground-based measurements.

Along the line of these recent research studies, in this work, an innovative cloud identification/classification technique, with application to infrared high spectral resolution synthetic data including the far infrared part of the spectrum, is presented. Many different cloud detection techniques exist and the majority of them exploit data spanning from infrared to shortwave, which limits their applicability to daytime hours only. An example of such a technique is the Principal Component Analysis (PCA) based detection algorithm presented by Ahmad (2012). Other techniques rely on outgoing longwave radiation only; one example is the algorithm presented in Serio et al. (2000), which is tailored to work on water surfaces. Among the techniques exploiting outgoing longwave radiation only, and applied globally, the cumulative discriminant analysis by Amato et al. (2014), the CNRM scheme by McNally and Watts (2003), the Met Office 1D-Var retrieval system (Pavelin et al. (2008)), the NCEP minimum residual method (Eyre (1989)), and the CMS cloud mask (Lavanant and Lee (2002)) are mentioned. These methodologies are applied to spaceborne spectrometers and radiometers such as AIRS, IASI, and AVHRR. The last three methodologies perform cloud detection and classification simultaneously. They also depend on ancillary information regarding the atmospheric state, derived from NWP models. McNally and Watts (2003), for example, analyse the difference between simulated clear sky spectra and the measured spectra to detect the presence of clouds, using the global model ECMWF to make a short term forecast of the atmospheric state.

Most cloud detection schemes rely on the definition of some statistical parameter, which serves as a classifier, and on a statistical technique used to assign a value to the classifier. Both the cumulative discriminant analysis and the Met Office 1D-Var retrieval system seek the minimisation of a cost function. This minimisation produces the classification label of the spectrum and an estimate of the cloud fraction, respectively.

Some of the detection algorithms used for cloud identification have been adapted for aerosol detection and some innovative methodologies have been recently developed. A comprehensive overview of the mostly widely used techniques for aerosol detection and classification, such as Feature detection methods based on thresholds and brightness temperature differences, Spectral Fitting minimisation methods, approaches based on look-up-tables and minimisation of the Mahalonobis distance, and methods based on singular value decomposition and PCA, is provided in Clarisse et al. (2013).





Taking the cue from existing cloud detection algorithms, a Cloud Identification and Classification (CIC) method is developed. CIC is a machine learning algorithm based on Principal Component Analysis, which performs an identification and classification using a single threshold applied to a univariate distribution named SID (see Section 3.2). It is, partially based on other works (see Malinowski (2002) and Turner et al. (2005) for reference) and, with respect to previously described methods,

has the advantages of being easy to implement, user-friendly, fast and efficient.

In this paper, the CIC algorithm is used to perform cloud detection on a synthetic dataset consisting of infrared spectra with wavenumbers ranging from 100 to 1600 $cm^{-1}$ and a nominal resolution equal to 0.3 $cm^{-1}$, created to simulate satellite measurements of the FORUM mission. Cloud detection is performed both using the MIR only or the full spectrum (FIR and

MIR), so that the detection performances could allow an evaluation of far-infrared channels information content in realistic conditions. The algorithm is applied to simulated FORUM measurements from different climatic areas in order to observe the influence of atmospheric profiles on detection scores.

This paper is organized as follows. In Section 2 the synthetic dataset is illustrated; Section 3 is dedicated to CIC algorithm

description and functionalities. Section 4 deals with results obtained from CIC application to FORUM synthetic data. A brief summary is drawn in Section 5.

## 2 Synthetic dataset

A widespread dataset of simulated radiances for multiple atmospheric conditions is computed in order to test the cloud identification and classification algorithms. The synthetic dataset is build using a chain of codes to perform accurate line-by-line

multiple scattering computations as represented in Fig. 1. The computational methodology is similar to the one described by Bozzo et al. (2010) and the radiative transfer equation is solved through an adding-doubling algorithm for a plane-parallel geometry and simulating the FORUM satellite nadir view.

  The line-by-line computations of layer spectral optical depths are performed using the Line-by-Line Radiative Transfer Model LbLRTM v2.7 (Clough et al. (2005)), whose inputs are atmospheric vertical profiles and spectroscopic gas properties.

This model includes a recently updated water vapour continuum parameterisation, MT_CKD (Mlawer et al. (2012)) version 3.0, and a consistent spectroscopic database, AER version 3.5, built from HITRAN2012 (Rothman et al. (2013)). The atmospheric thermodynamic vertical profiles and gas mixing ratios (such as those of $H_2O$, $CO_2$, $CH_4$, $O_3$ and for a total of 22 molecules) are derived from different sources. The first one is the ERA-Interim Reanalysis (Dee et al. (2011)), with a horizontal resolution equal to 0.75 degrees (approximately 80 km) and 60 vertical levels from the surface up to 0.1 hPa. This database, that provides

4 sets of data per day, is easily accessible online using the Web Application Programming Interface (Web-API). Profiles of temperature, pressure, specific humidity, ozone mixing ratio, and geopotential height (from which the surface height is computed) are derived from the Era-Interim. The daily January, April, July, and October 2016 ERA-Interim Reanalysis data are





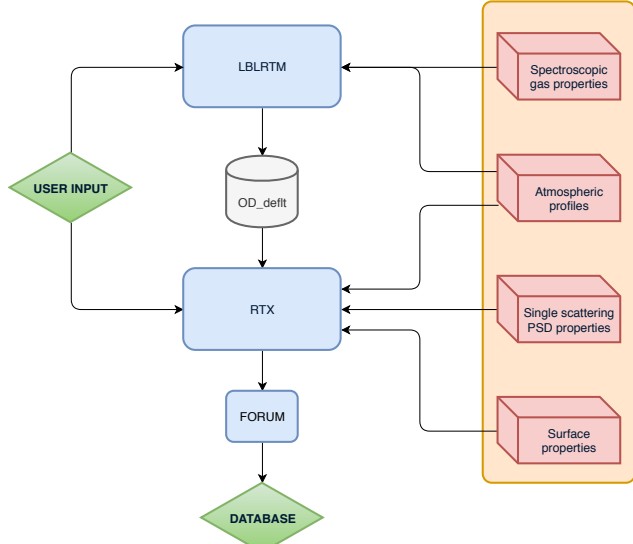

**Figure 1.** Software architecture (schematic) of the simulation process used to build the synthetic dataset for FORUM-like observations. Blue box are codes, red ones are auxiliary datasets. The FORUM box includes the Fourier Transform Spectroscopy and the simulation of the FORUM noise.

downloaded for the dataset generation in order to reproduce the seasonal and daily variations of the thermodynamic variables. Grid points from Tropics, Mid latitudes and Polar regions (Artic and Antarctic) are selected.

The second source of information is the Initial Guess climatological database IG2 (Remedios et al. (2007)) which includes atmospheric profiles for six latitude bands, four seasons, and two times of the day. This database spans from 2007 onwards,

with constant vertical resolution of 1 km, from the surface up to 120 km and provides time-averaged data concerning altitude, pressure, temperature and a wide range of gas mixing ratio profiles.

ERA-Interim Reanalysis data is thus used to characterise the daily variation of the main atmospheric parameters up to approximately 60 km of height and the IG2 is used to fulfil this information in the highest atmospheric layers and to provide information concerning minor gases mixing ratios.

Surface emissivity properties are selected in accordance with geolocation by using the global database produced by Huang et al. (2016). The database includes 11 types of spectral emissivity in the infrared region of the spectrum representatives of rainforest, temperate deciduous forest, conifer forest, grass, dry grass, desert, ocean, coarse/medium/fine snow, and ice.

The cloud microphysical properties are generated by integrating the Ping Yang database, consisting of single scattering properties of randomly oriented non-spherical ice crystals (Yang et al. (2013)), over a large set of gamma type size distributions.

Liquid water and mixed phase spherical particles radiative properties are derived through the Scattnlay code (Peña and Pal (2009)) and subsequently used to get bulk properties of particle size distributions. The mixed phase spheres consist of a core of ice surrounded by a coating of liquid water. This shell of liquid is modelled as a coating of 10 or 20% of the radius of the entire particle. In the simulations many cloud properties are varied: cloud top height, geometrical thickness, optical thickness,





**Table 1.** Range of variability of the main cloud properties used in the simulations. The habit type indicates the assumed pristine shapes of the ice crystals. The mixed phase water coating indicate the percentage of liquid water coating with respect to the dimension of the assumed spherical particle.

| Cloud properties | Range |
| --- | --- |
| Top height (km) | 1.0 – 17.0 |
| Geometrical Thickness (km) | 1.0 – 5.0 |
| Optical Depth | 0.02 – 30.0 |
| Habit Type | Plate, Column, |
| | Bullet rosette, Aggregate |
| Mixed phase water coating | 10% and 20% |
| Ice particle Eff. Dimension ($\mu$m) | 4 – 100 |
| Liquid water Eff. Radius ($\mu$m) | 3 – 15 |
| Mixed phase Eff. Radius ($\mu$m) | 3 – 15 |

particle size distribution, mean effective dimension, particle shape, and phase (ice, liquid water, and the two levels of mixed phase). The clouds properties input to the radiative transfer code are modified over large ranges of values in order to account for the largest possible variability encountered in nature. Some datasets and recent studies are considered as baseline. The International Satellite Cloud Climatology Project ISCCP (Rossow and Schiffer (1999), Hahn et al. (2001)) is an example.

Cirrus clouds properties in the Tropics and at Mid Latitudes are mostly based on what was found by Veglio and Maestri (2011) while Antarctic cloud properties are obtained from several sources (i.e. Adhikari et al. (2012), Bromwich et al. (2012), Lachlan-Cope (2010)). In Table 1 the range of variability of some key cloud properties are reported.

    As illustrated in Fig. 1, for each selected atmospheric condition, high spectral resolution optical depths of atmospheric layers are computed using LbLRTM and the results are passed as inputs to RTX and, when in presence of clouds, the gas

optical depths are merged with those derived from cloud properties. RTX, which is based on the doubling and adding method [Evans and Stephens (1991)] and thus capable to solve the full radiative transfer equation in multiple scattering conditions, is then run to obtain the high spectral resolution radiances that are finally convoluted with an ideal FORUM like Instrument Line Shape (ILS) that is assumed to be a sinc function.

    This operation produces unapodised spectra with 0.3 $\mathrm{cm}^{-1}$ spectral resolution over a spectral range spanning from 100 to

1600 $\mathrm{cm}^{-1}$ representative of FORUM mission noiseless observations. In a subsequent step, an ideal measurement noise is added to the simulated radiances in order to produce a realistic FORUM observations dataset. The new dataset is computed by adding a gaussian wavenumber-dependent noise to the noiseless radiances. The spectral dependence and amplitude of the noise are derived from the technical specification of the FTS instrument described in the FORUM proposal to ESA (available on request) and reported in Table 2.





**Table 2.** Random noise used to simulate the fourier transform spectrometer of the FORUM mission.

| Interval (cm$^{-1}$) | Noise (mW/(m$^2$srcm$^{-1}$)) |
|---|---|
| 100-200 | 1.0 |
| 200-800 | 0.4 |
| 800-1600 | 1.0 |

**Table 3.** Number of clear and cloudy simulated spectra for each latitude belt. In parenthesis the number of liquid water clouds (Liq), of mixed phase clouds (Mixed), of ice clouds (ice) and, among these, the number of sub visible cirrus (svc), consisting in high altitude cirri with optical depths less than 0.03, are shown.

| | Clear sky | Clouds [Liq / Mixed / Ice (svc)] |
|---|---|---|
| Tropics | 704 | 986 [16 / 61 / 909 (212)] |
| Mid Latitudes | 615 | 765 [96 / 48 / 621 (173)] |
| Polar regions | 492 | 532 [00 / 48 / 484 (132)] |

A spectral random noise is computed for each spectrum. The Central Limit theorem is used so that the sum of random numbers ($r_{tot}$) from a uniform distribution ranging from -0.5 to 0.5 (variance is 1/12) is used to generate a gaussian variable ($r_{gauss}$) with mean 0 and a standard deviation, $\sigma_\nu$, assumed to be equal to the FORUM noise. The spectral random noise is thus obtained by using the following formula:

$$r_{noise} = \sqrt{\frac{12}{r_{tot}}} \cdot \sigma_\nu \cdot r_{gauss}$$

A schematic summary of the whole dataset, comprising of 4244 simulated spectra, is presented in Table 3 for each latitude belt and for the clear or cloudy class. Some example of spectra are shown in Fig. 2, 3, 4 and 5.

In Fig. 2, the brightness temperature sensitivity (with respect to a reference clear sky case) is shown for three different cases in the Tropical atmosphere: a sub-visible cirrus cloud (black line), the clear sky case with a 3 K decrease in skin surface

temperature (yellow line) and the clear sky case with a an increase of 10% along the vertical profile of the water vapour mixing ratio (blue line). The cirrus is assumed to be composed of plates with effective dimension of 20 μm, optical depth OD=0.03 and geometrical thickness 1 km. Results show distinctive spectral features due to a presence of the cirrus cloud in the satellite view. In particular, for the tropical region, the radiance signal in the FIR exiting from the surface is masked by the strong absorption by the water vapour rotational band (which is almost saturated for wavenumbers below 300 cm$^{-1}$).

In Fig. 3 the large BT sensitivity to cloud particle size distribution effective dimensions is highlighted for channels in the MIR windows and also for wavenumbers between 400 and 600 cm$^{-1}$. Cloud properties are: OD=1.5, geometrical thickness 1 km, cloud top is at 14 km.

Sensitivity to cloud particle phase is shown in Fig. 4. In the computations a Polar cloud (OD=7, geometrical thickness 1.5 km, cloud top is at 5 km) made of spheres of pure ice, liquid water or of two diverse mixed phases (see figure caption for the





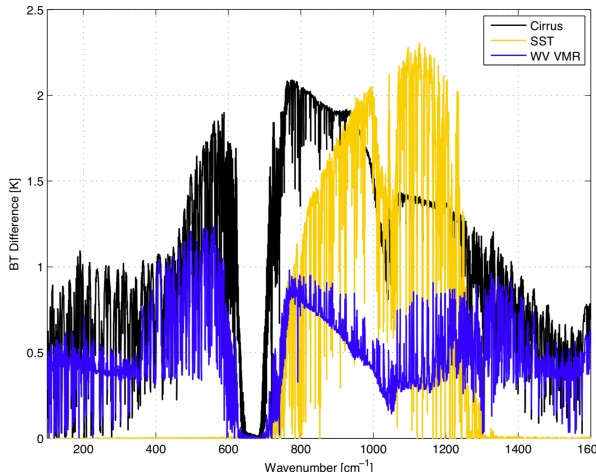

**Figure 2.** Brightness Temperature (BT) sensitivity to a sub-visible cirrus cloud (black line), to a 3 K decrease in surface temperature (yellow line) and to an increase of 10% in water vapor mixing ratio at all levels (blue line). The BT differences are obtained in reference to a Tropical clear sky case over the ocean.

details) is assumed. The results show that the highest BT sensitivity to phase is found at FIR wavenumbers. Note also how the mixed phased spheres resemble the pure ice phase in the MIR window channel and the pure liquid water in the FIR channels.

Finally, in Fig. 5 the sensitivity to crystal habit is shown. Four different habits (aggregates, plates, bullet rosettes and solid columns) are assumed in the simulation of 4 different tropical cirrus clouds with the same features (OD=1, geometrical thick-
5 ness is 2 km, cloud top is at 15 km and effective dimension is 32 μm). Habit sensitivity is much larger in the FIR (about 5 K spread among the curves for different shapes) than in the MIR windows (about 2 K). This is mostly due to a minimum in the imaginary part of the ice refractive index at around 410 cm$^{-1}$ that imply a minimum in absorption at FIR wavelengths and a relative larger importance of scattering processes that are related to crystal shape. This FIR largest sensitivity is noted to increase with the dryness of the atmosphere and thus amplified when moving towards higher latitudes (not shown here).

**3 Cloud Identification and Classification (CIC**

**3.1 Algorithm description**

CIC is an innovative classification algorithm based on Principal Component Analysis. The methodology relies on a machine learning algorithm that requires the definition of a certain number of training sets equal to the number of classes used for the classification. A descriptive example of the identification process of clear and cloudy cases (*cloud detection*) is first provided
in order to facilitate the comprehension of the rigorous mathematical treatment that follows this brief introduction. In Fig. 6 a flowchart of the algorithm is depicted.



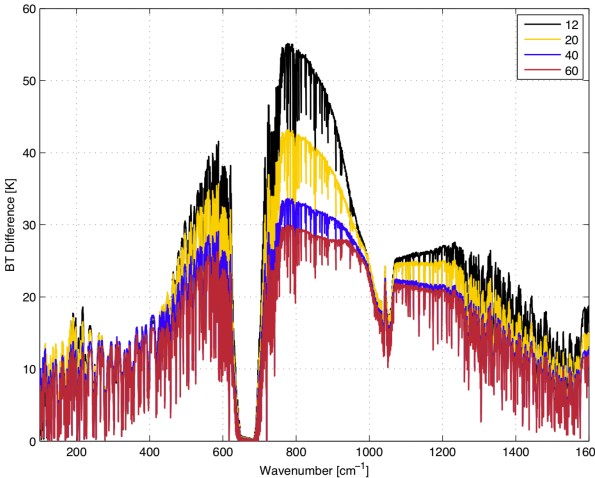

**Figure 3.** Brightness Temperature sensitivity to cirrus cloud particle size distribution effective dimensions. Dimensions values in the legend are in $\mu m$. The assumed shape is the plate type, optical depth is 1.5, the cloud thickness is 1 km and cloud top is at 14 km. The BT differences are obtained in reference to a Tropical clear sky case over the ocean.

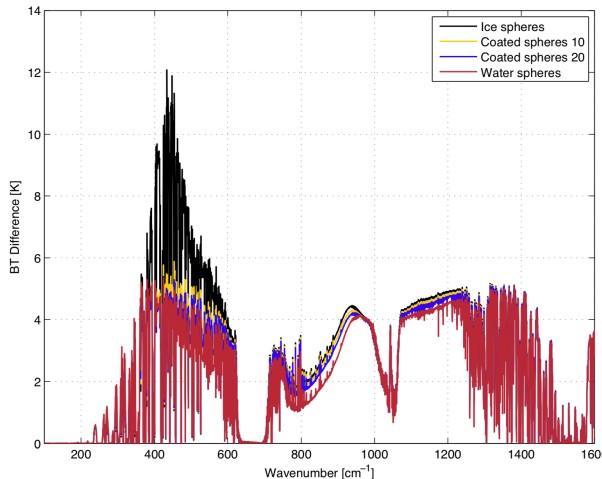

**Figure 4.** Brightness Temperature sensitivity to particles phase. A polar cloud is assumed made of a particle size distribution of spheres of ice (black line), liquid water (red) and two mixed phases. The mixed phase are spheres with an ice core and liquid water coating. The coating is respectively a 10% (yellow line) and a 20% (blue line) in radius of total sphere radius. Cloud optical depth is 7, cloud thickness is 1.5 km and cloud altitude is 5 km. The BT differences are obtained in reference to an Antarctic clear sky case over a snowed surface.

The first step requires the definition of a *clear sky training set* ($\mathbf{TR}_{\mathrm{cle}}$), consisting of a number $T_{cle}$ of clear sky spectra, and a *cloudy sky training set* ($\mathbf{TR}_{\mathrm{clo}}$), consisting of $T_{clo}$ cloudy sky spectra. For each training set the principal components





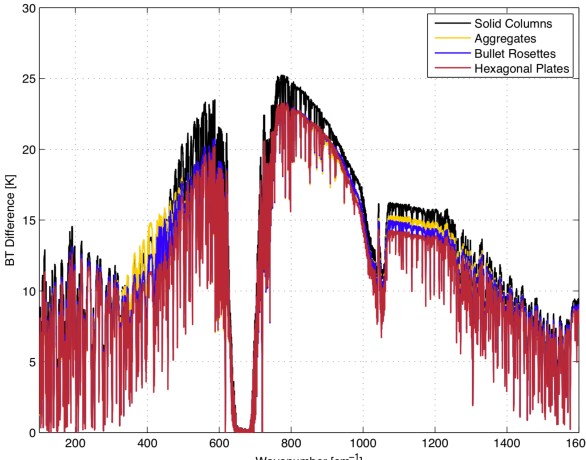

**Figure 5.** Brightness Temperature sensitivity to cirrus cloud crystal habit. The crystal's habits assumed are reported in the legend. The simulated tropical cirrus cloud is 2 km thick, with optical depth 1 and with effective dimension equal to 32 $\mu m$. The BT differences are obtained in reference to a Tropical clear sky case over the ocean.

(PCs) are computed and stored in a matrix. Each spectrum of the test set (one at a time) is then added to the $\mathbf{TR}_{\mathrm{cle}}$ and thus an *extended clear sky training set* ($\mathbf{ETR}_{\mathrm{cle}}$) is defined. $\mathbf{ETR}_{\mathrm{cle}}$ is a group of $T_{cle} + 1$ spectra. The principal components of the $\mathbf{ETR}_{\mathrm{cle}}$ are computed. Supposing that the test set spectrum under consideration is a clear sky spectrum it is expected that the PCs computed for $\mathbf{ETR}_{\mathrm{cle}}$ are very similar to the ones computed for $\mathbf{TR}_{\mathrm{cle}}$ that is to say that the test set element has the same

basic features as the elements belonging to the training set (clear in the running example). In this case, it is also expected that the PCs computed for the cloudy sky training set $\mathbf{TR}_{\mathrm{clo}}$ differ from the PCs obtained from the extended cloudy sky training set ($\mathbf{ETR}_{\mathrm{clo}}$) that is obtained by adding the spectrum in consideration (that is clear) to the cloudy sky training set.

CIC evaluates the variation of the principal components of the training sets due to the addition of a new spectrum (from the test set). The association of a spectrum to a specific class is obtained by evaluating the similarity of PCs of the extended

training sets to those of the original training sets: *small* changes in PCs are interpreted as that the spectrum belongs to the class while large changes suggest that the spectrum belongs to a different class. The variations of the PCs obtained for the extended training sets with respect to the original ones are evaluated by means of a new parameter called *similarity index*.

The notation for similarity indices is:

$$\mathrm{SI}(i,j), \qquad i \in \{1,2\}, \quad j \in \{1...J\} \tag{1}$$

where $i$ is the class label, $j$ is the test set spectrum label and $J$ is the number of spectra in the test set to be classified. As an example (that is used in the whole text) it is assumed that the class label is 1 for clear sky spectra and 2 for cloudy sky spectra.





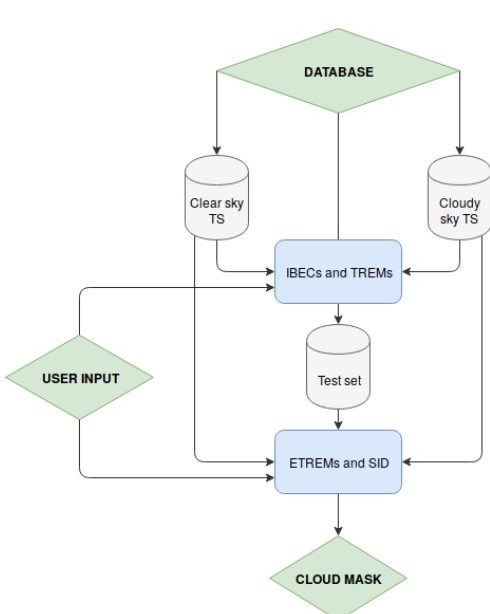

**Figure 6.** Scheme of the dataflow used for the cloud detection process.

The computation of the similarity indices are now described mathematically. The first step is the definition of the *training set matrices*.

$$\mathbf{TR}_i(\nu, t) \tag{2}$$

$$i \in \{1, 2\}, \quad t \in \{1...T_i\}, \quad \nu \in \{1...\nu_{max}\}$$

where $t$ is the spectrum index, $T_i$ is the number of spectra in each training set $i$, $\nu$ is the wavenumber index that spans from 1 to $\nu_{max}$ that is the highest wavenumber index.

The second step consists in the computation of the PCs of each training set matrix by evaluating the eigenvectors (eig) of their covariance (cov) matrix:

$$\mathbf{TREM}_i(\nu, p) = \text{eig}(\text{cov}(\mathbf{TR}_i(\nu, t))) \tag{3}$$





$$i \in \{1,2\}, \ t \in \{1...T_i\}, \nu \in \{1...\nu_{max}\}, \ p \in \{1...P\}$$

where $\mathbf{TREM}_i$ is the *training eigenvector matrix*, $p$ indicates the $p^{th}$ principal component, and $P = \max(T_i, \nu_{max})$ is the total number of principal components. Each line of this matrix contains normalised eigenvectors:

$$\sum_{\nu=1}^{\nu_{max}} \mathbf{TREM}_i(\nu,p)^2 = 1 \qquad (4)$$

Given $J$ spectra in the test set, a number of $J$ new matrices is defined for each class. These matrices are simply the concatenation of the *training set matrices* with each single spectrum ($j$) from the test set, and are called *extended training set matrices*. Let

$$\mathbf{TS}(\nu,j), \quad 1 < j < J$$

be the matrix containing all the spectra from the test set. The *extended training set matrices* are defined as follows:

$$\mathbf{ETR}_{i,j}(\nu,t') = [\mathbf{TR}_i(\nu,t)\|\text{row}_j(\mathbf{TS}(\nu,j))] \qquad (5)$$

$$t' \in \{1...T_i+1\}$$

where the notation $\|$ indicates matrix concatenation. Note that $\text{row}_j(\mathbf{TS}(\nu,j))$ is a single test set spectrum in a one-dimensional array.

CIC evaluates the principal components of the *extended training set* $\mathbf{ETREM}_{i,j}$ as follows:

$$\mathbf{ETREM}_{i,j}(\nu,p) = \text{eig}(\text{cov}(\mathbf{ETR}_{i,j}(\nu,t))) \qquad (6)$$

The training and extended training eigenvector matrices are used to compute the similarity indices (SI) for each test set spectrum ($j$) and for each class ($i$):

$$\text{SI}(i,j) =$$

$$1 - \frac{1}{2P_o} \sum_{p=1}^{P_o} \sum_{\nu=1}^{\nu_{tot}} |(\mathbf{ETREM}_{i,j}(\nu,p))^2 - \mathbf{TREM}_{i,j}(\nu,p)^2| \qquad (7)$$

where $\nu_{tot}$ is the number of features (channels) used for PCA analysis and $P_o$ is the number of principal components that are associated to the physical signal (real variability) characterizing the spectrum.

The set of optimal principal components ($P_o$) characterizing the signal constitutes the Information BEaring Principal Components (IBECs). The $P_o$ elements are extracted by minimizing the factor indicator function (IND) defined by Malinowski (2002) and Turner et al. (2005):

$$\text{IND}(p) = \frac{\text{RE}(p)}{(P-p)^2} \qquad (8)$$

where $\text{RE}(p)$ is defined, in Turner et al. (2005), as the *real error*

$$\text{RE}(p) = \sqrt{\frac{\sum_{i=p+1}^{P} \lambda_i}{T_i(P-p)}} \qquad (9)$$





where $\lambda_i$ is the $i^{th}$ eigenvalue of the covariance of some data matrix and $T_i$ is the number of spectra in the training set $i$.

The natural number $P_0$, obtained through this minimisation process, is the number of eigenvectors associated with the physical signal corresponding to the number of IBECs. In CIC, $P_o$ is computed separately for both training set matrices ($i = 1, 2$).

Once $P_o$ is determined the similarity index can be calculated using equation 7. For consistency, the same value of $P_o$ is used when the $SI$ computation is applied to the two training sets; the minimum value for $P_o$ is utilised.

Interpreting the eigenvectors as directions in the multi-dimensional space, SI is an estimate of how much the principal components of the training set *rotate* after a new spectrum is added to the set. For this reason, similarity indices do not depend on *eigenvalues* but on *eigenvectors*: all the principal components describing the physical signal are accounted for with the same weight in equation 7, since all of them might be important for classification.

Similarity indices defined in this way are normalised. In fact, since the absolute value of the difference between the square loadings of two eigenvectors is at most equal to 2, the sum of $P_o$ differences can reach the maximum value of $2P_o$. And being an absolute value, it turns out that:

$$0 \leq \sum_{p=1}^{P_o} \sum_{\nu=1}^{\nu_{tot}} |(\mathbf{ETREM}_{i,j}(\nu, p))^2 - \mathbf{TREM}_i(\nu, p)^2| \leq 2P_o \tag{10}$$

and therefore:

$$0 \leq \mathrm{SI}(i, j) \leq 1 \tag{11}$$

With reference to equation 7, the largest value of the similarity index (1) is obtained for identical TREM and ETREM matrices meaning that the analysed test set spectrum is not adding any diverse physical information to the training set spectra. An SI close to zero means that the two matrices are described by very different PCA and the test set element is bringing additional information to the original training set.

A graphical example of the similarity index is provided in Fig 7. The plot shows the SI computed for cloudy elements of the tropical test set only in the left hand side panel and for clear sky elements only in the right hand side panel. For cloudy sky test set cases, when the SI is computed using the cloudy training set (orange line) the SI is very close to one while the SI values are mostly lower when the clear sky training set is used. Thus, the inequality $\mathrm{SI}(clo, j) > \mathrm{SI}(cle, j)$ holds for most cloudy sky spectra $j$. The situation is reversed when clear sky elements of the test set are used (left panel) showing that highest SI values are generally obtained when using the clear sky training set (blue line). CIC exploits results from both the comparisons: SI computed using the clear sky and the cloudy sky training sets.



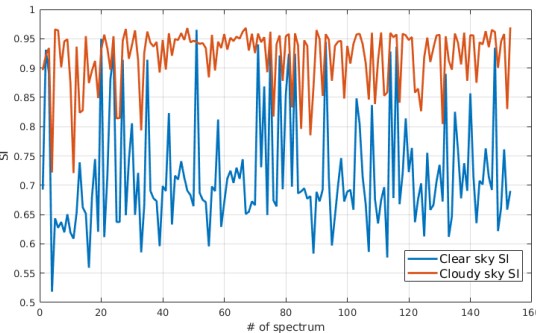
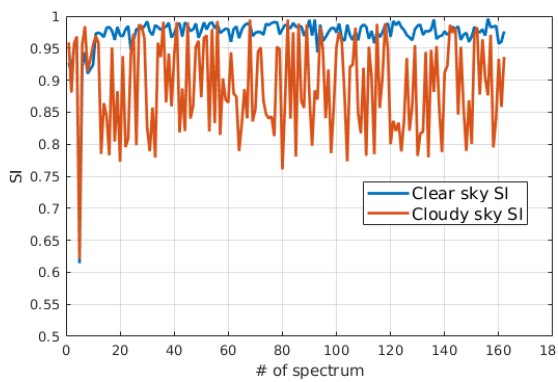

**Figure 7.** Similarity indices computed using tropical cloudy spectra only (left panel) or clear sky spectra only (right panel). In orange the SI is computed using the cloudy sky training set and in blue the clear sky training set. The tropical case is accounted for. The full spectrum is used in the SI computations. Details on the training sets are provided below in the text.

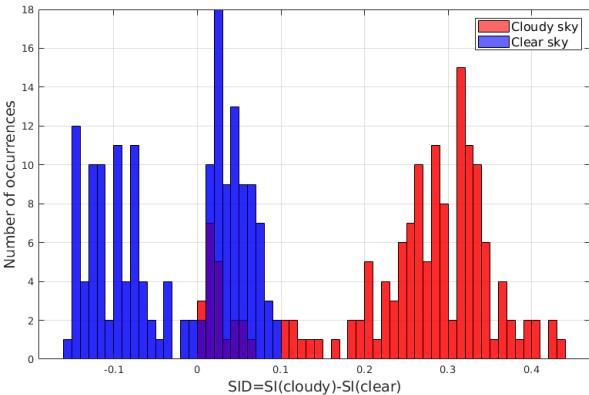

**Figure 8.** SID occurrence distribution for a subset of tropical spectra. The SIDs are computed based on specific (not optimised) training sets. Test set clear sky spectra are in blue, while cloudy sky spectra are in red. Cirrus clouds show very small, positive values of the SID, comparable with values obtained for some clear sky spectra. Class membership of test set spectra is known a-priori since the dataset is synthetic.

## 3.2 Classification

### 3.2.1 Elementary approach

CIC classification requires that each test set element is used for the computation of both similarity indices (one for each training set of the two classes). Once the SIs for the extended training sets (the one containing cloudy spectra and the one for clear





ones) are computed a comparison is performed. Continuing with the running example, when

$$\mathrm{SI}(\mathrm{cle}, j) > \mathrm{SI}(\mathrm{clo}, j),$$

then spectrum $j$ is expected to be clear sky (cle). And of course for:

$$\mathrm{SI}(\mathrm{clo}, j) > \mathrm{SI}(\mathrm{cle}, j)$$

the spectrum $j$ is expected to be cloudy (clo).

These two conditions may be unified in a compact index that is defined as the Similarity Index Difference (SID):

$$\mathrm{SID}(j) = \mathrm{SI}(\mathrm{clo}, j) - \mathrm{SI}(\mathrm{cle}, j)$$

and thus if

$$\text{if } \mathrm{SID}(j) > 0, \quad \text{then } j \in \{\text{cloudy spectra}\} \tag{12}$$

$$\text{if } \mathrm{SID}(j) < 0, \quad \text{then } j \in \{\text{clear spectra}\} \tag{13}$$

Spectra classification based only on the SID sign is defined as the *elementary approach*: SID acts as a binary classification parameter. Each spectrum is analysed sequentially and independently from the other elements of the test set under consideration. This elementary approach has the main advantage of being very simple and straightforward, and the disadvantage of being sensitive to the composition of the training sets. In fact, results might be affected (and biased) if one of the training

sets is not well populated by spectra that are representatives of the variability within the class. An example of the elementary classification is given in Fig. 8 where the SID distribution for clear and cloudy sky spectra is shown. In the example, the cloudy sky and clear sky training sets are not well characterised (some unbalance is observed) since even if all the cloudy spectra show positive SIDs (as expected), a large number of clear sky spectra has also positive SIDs and thus, those spectra, are potentially misclassified. It is therefore shown that the elementary approach requires an accurate definition of the training sets to work

properly.

### 3.2.2   Distributional approach

A *distributional approach* can be adopted for the classification in which the distribution of the SIDs of the training set is analyzed before performing the classification.

A perfect classifier would ideally generate a bimodal SID distribution. Thus the transition from one mode to the other could be associated to a binary classification parameter that changes sign in the transition point. In order to accomplish this task a new index is defined called the *corrected similarity index difference* (CSID).

The elements of the training set (both the clear and cloudy spectra) are used to mathematically define the CSID (that is simply a shifted SID):

$$\mathrm{CSID}(j) = \mathrm{SID}(j) - shift_{opt} \tag{14}$$





where $shift_{opt}$ is the optimal value of the parameter *shift* that maximises a function that can potentially forecast the performance of the algorithm (a *performance forecasting function*, PFF). This function is called Consistency Index (CoI) and is defined below:

$$\text{CoI}(\text{shift}) = 1 - \max(\frac{\text{FP}_{\text{cle}}}{T_{cle}}, \frac{\text{FP}_{\text{clo}}}{T_{clo}}) \tag{15}$$

$$\tag{16}$$

where the clear and cloudy false positives (FP) are counted as:

$$\text{FP}_{\text{cle}}(\text{shift}) = \text{card}(\{t \in T_{cle} \mid \text{SID}(j) - \text{shift} > 0\}) \tag{17}$$

$$\text{FP}_{\text{clo}}(\text{shift}) = \text{card}(\{t \in T_{clo} \mid \text{SID}(j) - \text{shift} < 0\}) \tag{18}$$

In the above formula, the «card» operator denotes set cardinality, while $T_{cle}$ and $T_{clo}$ are the number of clear and cloudy training set elements respectively. The consistency index measures the representativeness of the training sets and in particular it computes how many training set elements would be classified correctly if they were part of the test set. The consistency index is large (close to 1) only if both clear and cloudy false positives (FP in the equations above) are rare, that is when the training sets are composed of elements that accurately represent the variability within the specific class.

The new CSID parameter operates as the SID parameter: if it is positive, the spectrum is considered cloudy, and if its negative, the spectrum is considered clear. In formulas:

$$\text{if } \text{CSID}(j) > 0, \quad \text{then } j \in \{\text{cloudy spectra}\} \tag{19}$$

$$\text{if } \text{CSID}(j) < 0, \quad \text{then } j \in \{\text{clear spectra}\} \tag{20}$$

The use of the distributional approach significantly improves the performance of the algorithm as it will be shown later in the text.

### 3.3 Unclassified spectra

Each element of the test set is classified in accordance with the sign of the classifier (SID or CSID). For practical purposes, it can be useful to define some thresholds or limits that determine a set of «unclassified» elements characterised by values of the classifier belonging to a limited interval around 0 (that is the ideal point separating the two modes of the distribution). The simplest way to set this interval is to let the user to define two parameters: $\Theta_1$ (positive) and $\Theta_2$ (negative), representing the inner limits of a confidence interval. Any spectrum whose classifier is such that falls within the interval $[\Theta_2, \Theta_1]$ is considered «unclassified». The $\Theta_2$ and $\Theta_1$ parameters should be defined in accordance with the experimental conditions and sensor performances and their quantification is beyond the scope of this work. For this reason and since we rely on a synthetic test set all the classifications performed will be binary that is each element is classified either clear or cloudy.





### 3.4  Scores

There is no unique metric to define the performances of an algorithm since their assessment is linked to the goal of the study. For this reason, the set of parameters (i.e. scores) used to evaluate the algorithm performance in this section are somewhat arbitrary. As a general rule, any metric should measure a better performance if more spectra are classified correctly.

At this regard, an element of the test set that undergoes a classification process falls into one of the following categories:

1.  *True Positive. TP*: the element is a class $i$ member, and is correctly classified.

2.  *False Positive, FP*: the element is not a class $i$ member, but it is classified as belonging to class $i$.

3.  *False Negative, FN*: the element is a class $i$ member, but is not correctly classified as such.

4.  *True Negative, TN*: the element is not a class $i$ member, and it is not classified as belonging to class $i$.

It is possible to define the classification performance using two or more of these sets.

A possible definition of performance is provided by the *a-priori* classification score, here called PRISCO:

$$\text{PRISCO}(i) = \frac{\text{TP}(i)}{\text{TP}(i) + \text{FP}(i)} \tag{21}$$

Where $\text{TP}(i)$ is the number of class $i$ true positives and $\text{FP}(i)$ is the number of class $i$ false positives. The PRISCO is known as
*hit rate* in the literature (see Wilks (2006)). This score ranges between 0 and 1. It is maximised for $\text{FP}(i) = 0$, occurring when all the spectra classified by the algorithm as class $i$ are actually belonging to class $i$.

This score can be viewed as an *a priori probability*, i.e. the likelihood that a spectrum *labelled* as a member of class $i$ actually belongs to class $i$,:

$$\text{PRISCO}(i) = \text{P}(\text{L}(i)|\text{B}(i)) \tag{22}$$

where $L(i) = \frac{\text{TP}(i) + \text{FP}(i)}{\text{CTE}}$ is the fraction of spectra *labelled* as class $i$, and $B(i) = \frac{\text{TP}(i) + \text{FN}(i)}{\text{CTE}}$ is the fraction of spectra *belonging to* class $i$. The variable CTE represents the cardinality of the test set elements.

The *a-posteriori* probability, i.e. the likelihood that a spectrum belonging to class $i$ is correctly classified by the algorithm,
is defined by the classification score here called POSCO:

$$\text{POSCO}(i) = \frac{\text{TP}(i)}{\text{TP}(i) + \text{FN}(i)} = \text{P}(\text{B}(i)|\text{L}(i)) \tag{23}$$

where $\text{FN}(i)$ is the number of class $i$ false negatives. This score is useful to estimate the percentage of correctly classified spectra per each class: in fact, the sum of class $i$ true positives and false negatives ($TP(i) + FN(i)$) is equal to the total number





of class $i$ spectra.

The two scores are complementary and are related by the equation:

$$\text{POSCO}(i)B(i) = \text{PRISCO}(i)L(i) \tag{24}$$

In this paper, detection performance (DP) is defined as the minimum value out of the two PRISCO scores relative to the classes under examination:

$$\text{DP} = \min(\text{PRISCO}(\text{class 1}), \text{PRISCO}(\text{class 2})) \tag{25}$$

where:

$$0 \leq \text{DP} \leq 1 \tag{26}$$

The PRISCO is preferred to POSCO since no a-priori assumption can be made about class membership when classification is performed on real data. In the definition of the detection performance the minimum PRISCO value is chosen because the classification performance is considered high only if hit rates for both classes are high.

## 4    Results

In this section the performance of the CIC cloud detection algorithm are evaluated for multiple atmospheric conditions. The additional information content of the FIR part of the spectrum is also studied. Classifications are performed both by using the mid infrared (MIR) channels only or by using channels spanning over the full FORUM spectrum (FIR+MIR).

A first result is anticipated in Fig. 9; the details of the computations will be described in the next paragraph. Multiple couples of training sets (clear and cloudy), 100 elements in total for each couple, are used to performed a classification applied to the simulations for the tropical case. For each training set (60 in the example) CIC is applied, the Consistency Index and the detection performance are computed and plotted as scatter plots. This allows to relate the final classification scores (evaluated by DP) with the composition of the training set (whose characterisation is associated to the CoI). The CIC algorithm is run using both the MIR only or the full spectrum: 129 channel in the FIR and 129 in the MIR. The selected channels range in the $[371.1 - 1300]\,\text{cm}^{-1}$ interval and are selected with a fixed sampling but a small interval at around the $667\,\text{cm}^{-1}$ $\nu_2$ vibrational $CO_2$ band that is not used. Results show that the scores (indicated by DP) are generally larger when the full spectrum is exploited. In fact, it is computed that the average values of the DP move from 0,60 to 0,79 (for the elementary approach, left panel in the figure) and from 0.67 to 0.86 (for the distributional approach, right panel in the figure) when using features from the full FORUM spectra instead that those from the MIR only.

These preliminary computations suggest the following:

1. better results are obtained for the distributional approach (right panel) with respect to the elementary one (left panel)





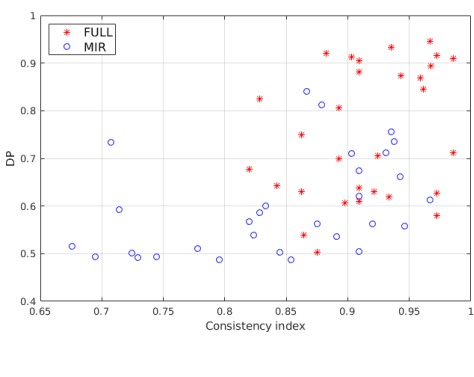 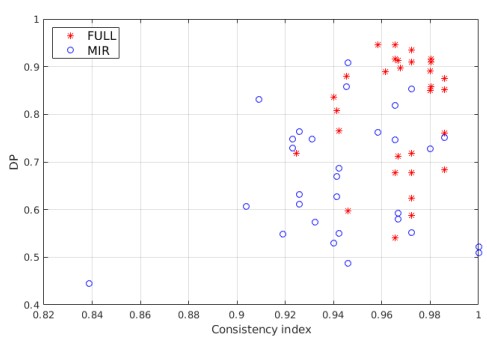

(a)                                              (b)

**Figure 9.** Scatter plots relating the consistency index (CoI) and detection performance (DP) for multiple tropical training set couples (clear and cloudy). CIC results are shown for both the elementary (left panel) and the distributional approach (right panel). Blue circles represent classification results obtained using features in the MIR only, while red stars represent classification results exploiting the full spectrum.

    2. a correlation between the CoI and the DP exists.

The points above are taken into account in order to maximise the performances of the CIC algorithm when applied to a test set. In particular, it is suggested that the distributional approach is preferred over the elementary one and that an optimal training set can be arranged before the classification is actually performed since CoI is computed for training set elements.

Thus, a strategy for the application of CIC to the synthetic dataset is defined and the evaluation of the information content in the FIR is planned:

    – A REference Training Set (called RETS and consisting of a clear and a cloudy training set) is defined using an optimisation methodology based on the CoI values. The RETS is used as a reference to perform test set classification with a variable number of features.

– The performance of the algorithm is studied when using features from the MIR only or when exploiting also an increasing number of FIR channels up to cover the full FORUM spectrum.

## 4.1  REference Training Set (RETS)

In this paragraph, a strategy for the definition of an optimised training set, *RETS*, (intended as a set of clear and a set of cloudy spectra) is outlined.

The strategy is based on the correlation between CoI and the CIC performances (measured by the DP parameter). As said, the CoI measures how well the training set elements would be classified if they were part of the test set.





For simplicity, only the CIC applications to the tropical data set are reported in this section. The Tropical dataset includes 352 clear sky and 817 cloudy sky spectra, for a total of 1169 spectra. Out of these, 315 are used as test set. The other spectra are exploited to define multiple training sets: 100 elements at a time for each training set couple (clear and cloudy). This strategy is followed in order to have multiple training sets with different combinations of spectra. Of course the optimisation process

is particularly important when the spectra that need to be classified are many. The operation is also important in reducing computational cost (see Subsection 4.4). In fact, the TREM matrices and the SID distribution of the optimised training set can be saved in a file, reducing the running time of both the elementary and the distributional approach.

The training set elements (100 per each training set couple) are randomly chosen from a set of 854 simulations in clear

and cloudy sky, but with a constraint on the number of clear and cloudy components. Three different Training set Number Configurations (TraNCs) are used:

$$T_{cle}(1) = 70, \ T_{clo}(1) = 30 \tag{27}$$

$$T_{cle}(2) = 50, \ T_{clo}(2) = 50 \tag{28}$$

$$T_{cle}(3) = 30, \ T_{clo}(3) = 70 \tag{29}$$

Twenty different training sets are constructed for each TraNC. Both an elementary and a distributional-approach-based classification is performed for each training set.

CIC is run on the test set elements and a binary classification (each element is classified clear or cloudy) is performed by exploiting the full spectrum. The the scores (and DP) are computed. The results are presented in Fig. 10 where the DP is plotted

as a function of the CoI.

On average, the most accurate performances are obtained for TraNCs in which $T_{cle}$ is larger than $T_{clo}$. The worst results are obtained for $T_{clo}$ greater than $T_{cle}$. The average DP values for the three groups of TraNCs are reported in the plots of the figure.

Fig. 10 shows that a correlation exists between the DP and CoI in the sense that, on average, large values of DP are obtained

for large values of CoI. It is also shown that, on average, the DP computed using the distributional approach is larger than for results obtained using an elementary approach.

This correlation is significant because it is observed even when CIC computes these two parameters totally independently, i.e., when the elementary approach is followed. For this reason the CoI can be used as a *performance forecasting parameter*, that is, a parameter estimating the quality of the classification. Nevertheless a quantification of the performances cannot be

provided a-priori of the application of the algorithm since it also depend on the spectra of the test set to be analysed.

If a positive correlation between the DP and the CoI is assumed to exist, then the best performances are expected for training sets with the highest CoIs. For this reason, the RETS is defined as the training set with the highest CoI among those considered.





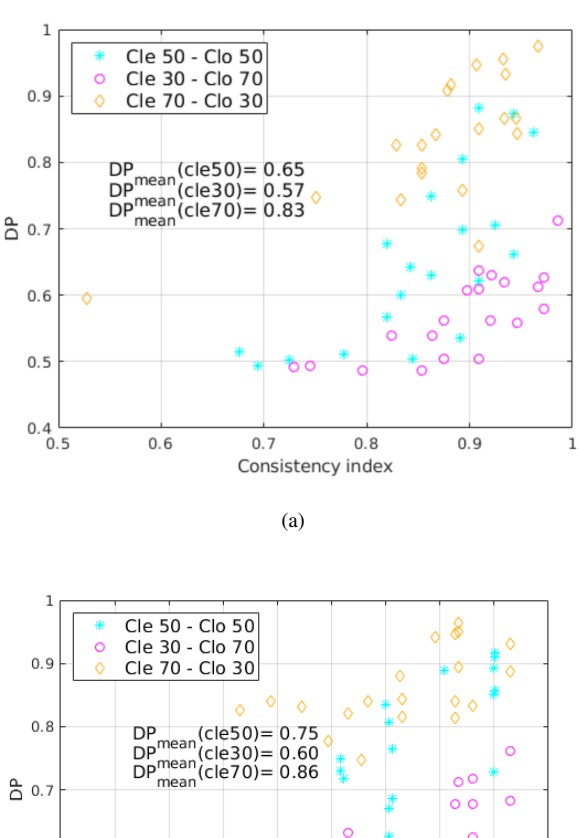

(a)

(b)

**Figure 10.** Scatter plot of DP and CoI for multiple training sets (grouped in 3 TraNC reported in the legend). Results are shown when using both the elementary approach (upper panel) and the distributional approach (lower panel). For each group an average value of the DP is computed and reported in the plot.

In our case, the best performing training set is composed of 70 clear sky and 30 cloudy sky spectra.

The above configuration is selected as the tropical RETS to be used in the analysis shown in the next paragraphs. Similarly RETS for the mid and the polar latitudes are constructed. The RETS for clear sky consists of a set of spectra able to catch the seasonal (and thermal) variability reported in the synthetic dataset which is created to reproduce global conditions for all the four seasons.

The cloud ODs accounted for in the cloudy RETS for Tropics range from 0.05 to 21.8, while the test set ones range from 0.02





**Table 4.** Number of cloudy spectra, as a function of cloud OD interval, for elements belonging to the cloudy set of the RETS and to the test set, Tropical case. In the second column, in parenthesis, it is reported the percentage value with respect to the total number of elements in the test set for the same OD interval.

| Training and test set cloud ODs | | |
|---|---|---|
| Range | RETS | Test set |
| OD $\leq 0.1$ | 8 (6.3%) | 126 |
| $0.1 < \text{OD} \leq 0.5$ | 3 (2.3%) | 131 |
| $0.5 < \text{OD} \leq 1$ | 0 (0%) | 19 |
| $1 < \text{OD} \leq 3$ | 11 (3.6%) | 308 |
| $3 < \text{OD} \leq 10$ | 0 (0%) | 21 |
| OD $> 10$ | 3 (5.0%) | 59 |
| Total | 30 (4.5%) | 664 |

to 23; the RETS's ice cloud top heights range from 9 to 16 km and in the test set they span from 4 to 16 km. A summary of the clouds OD used in the RETS, for the Tropical case, is reported in Table 4. It is shown that a large number of elements consists of optically thin clouds. This choice was performed in order to challenge the CIC capabilities to detect clouds in very difficult conditions.

## 4.2 Evaluation of FIR contribution to cloud identification

Multiple classifications, using a variable number of features (FORUM channels), and accounting both for the full spectrum and the MIR only are performed. The classifications account for a fixed number of MIR channels, while the number of FIR channels is changing to assess if the FIR part of the spectrum is capable to bring additional information content that significantly improves the algorithm's performances.

In order to speed up calculation and to avoid channels with a low signal to noise ratio, the chosen MIR wavenumbers only range from 667 to 1300 cm$^{-1}$, while the FIR ranges from 100 to 640 cm$^{-1}$. Thus the full spectrum spans over the 100-1300 cm$^{-1}$ spectral range with the exception of a limited wavenumber interval in the $CO_2$ band.

Channels are selected by using a constant sampling with no optimisation criterium applied. The number of selected channels in the FIR is defined by the following formula:

$$N_{feat} = \text{floor}(8 \cdot 2^{\frac{n}{2}}) + 1, \quad n \in 1...10 \tag{30}$$

$$\tag{31}$$

In this way, $N_{feat}$ spans over two orders of magnitude.





**Table 5.** Lowest wavenumber as a function of the number of FIR channels. The sampling rate is constant and set equal to 2.1 cm$^{-1}$. The highest FIR wavenumber is equal to 639.9 cm$^{-1}$.

| number of FIR channels | Lowest channel (cm$^{-1}$) |
| --- | --- |
| 12 | 616.8 |
| 17 | 606.3 |
| 23 | 593.7 |
| 33 | 572.7 |
| 46 | 545.4 |
| 65 | 505.5 |
| 91 | 450.9 |
| 129 | 371.1 |
| 182 | 259.8 |
| 257 | 102.3 |

In Table 5 the number of features used in the FIR are reported. The upper (starting) channel in the FIR is at 639.9 cm$^{-1}$ and the other FIR channels are sampled toward smaller wavenumbers every 2.1 cm$^{-1}$. Thus the data reported in the table should be interpreted as follows: 12 channels means that 12 channels between 639.9 and 616.8 cm$^{-1}$ are accounted for, and so on with the other larger number of channels up to cover the full FIR part of the spectrum.

In Fig. 11, the results obtained for 11 different classifications are shown in terms of detection performance. DP is plotted as a function of the number of FIR features used in the classification applied to the tropical case. At the value 0, of the x-axis of the figure, the MIR part of the spectrum only is accounted for (256 channels in this case), while in the other 10 cases the FIR part of the spectrum also is exploited with an increasing number of channels indicated by the x-axis values.

10 Results show that performance gradually improves for increasing number of FIR channels. In particular, there is a slight decrease in performance after adding the channels closest to the $CO_2$ $\nu_2$ band centre, which is gradually offset by improvements obtained when channels in the $[238.8 - 545.4]$ cm$^{-1}$ wavenumber range are added. The DP slightly decreses if channels in the $[102.3 - 238.8]$ cm$^{-1}$ range are included probably due to a reduced radiance sensitivity to surface and cloud features at those wavenumbers.

15 In the classifications, both the *distributional approach* (black line in the figure) and the *elementary approach* (red line) are followed. For the Tropical case, the elementary and distributional approach provide DPs as high as 0.9. Note that both methodologies take advantage of the optimised selection of the RETS and that the information content critical for DP improvements derives from channels spanning the $[238.8 - 545.4]$ cm$^{-1}$ range.





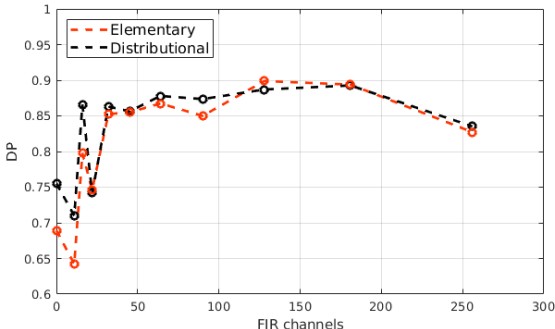

**Figure 11.** CIC cloud detection performance as a function of the number of FIR features (channels) for the Tropical case. 256 MIR channels are used. The black and red dots indicate the distributional and elementary approach respectively.

In Fig. 12 the cloud detection of the Tropical test set is plotted as function of the CSID value and cloud optical depth. Two classifications are performed: one using the MIR (left panel of the figure) and another using the full spectrum (right panel). The number of MIR channels used in the process is kept the same (128) for both configurations. The considered FIR channels, when using the full spectrum, are 129 and span from 371.1 to 639.9 cm$^{-1}$. For visual purposes the plotted clear sky test set

elements are associated to OD=$10^{-8}$, while all the other OD values are for cloudy sky. Results shown in Fig. 12 are obtained using the RETS.

True positives (correct classification) for clear and cloudy spectra are respectively orange circles and red asterisks. False positives are blue circle for clear and green asterisks for clouds. Left plot of the figure shows that the number of misclassified cloudy spectra grows for decreasing optical depth when using the MIR only. Clear sky case are all well classified in this

configuration. If CIC is run exploiting the full spectrum (right panel of the figure) the overall detection performance is enhanced even if a clear sky case is misclassified (green asterisk). Nevertheless, most cirrus clouds are now correctly classified, with the exception of only few cases with optical depth less than 0.5. Note that the DP value is the minimum between the hit rate computed for cloudy sky cases and clear sky cases.

### 4.3   Mid Latitudes and Polar regions

The CIC code is applied to the Mid Latitudes and Polar datasets to test the algorithm performances in different atmospheric conditions. The results obtained using the MIR only and using the full spectrum are again compared. Since detection performances are dependent on the analysed datasets (test sets) the results cannot be interpreted in absolute sense, but in reference to the configuration parameters.

Mid Latitude winter (MIDWIN), Polar winter (POLWIN) and Polar summer (POLSUM) cases are presented. The TraNCs used are generated with the same methodology outlined for the Tropical case. In Table 6 a summary of the total number of





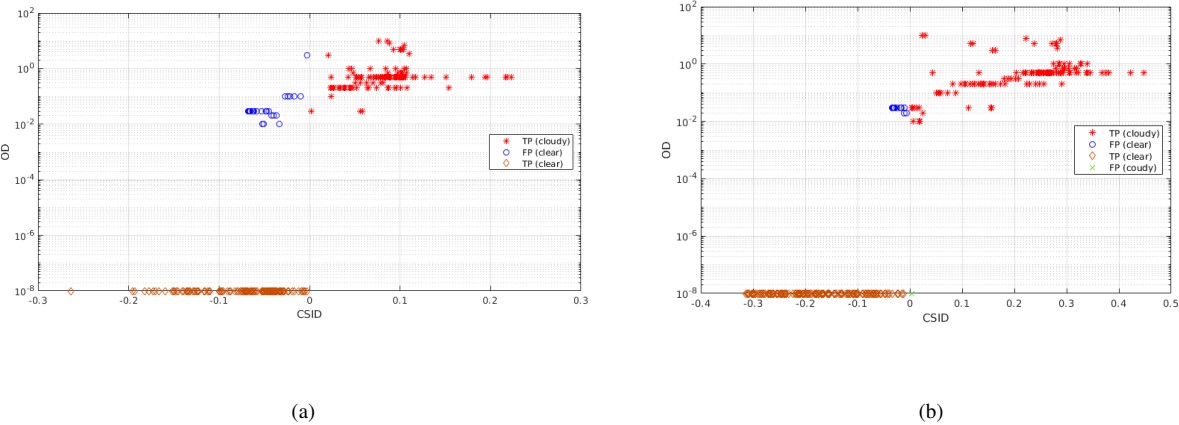

|  | (a) | (b) |

**Figure 12.** Classification of the Tropical test set as a function of the optical depth of the elements (a value of $10^{-8}$ is used to plot the clear sky cases). Data are plotted as a function of the corrected similarity index difference (CSID) and cloud optical depth (OD). Left panel: CIC is run using 128 MIR channels. Right panel: CIC is run using the 128 MIR and 129 FIR channels. For both cases the *distributional* approach is assumed. Color codes are reported in the legend.

**Table 6.** MIDWIN, POLWIN, and POLSUM data set. The number of simulations used to define the training sets and the number of cases in the test sets for each region are reported.

|  | MIDWIN | POLWIN | POLSUM |
|---|---|---|---|
| Clear | 305 | 244 | 248 |
| Train. Set clear | 205 | 164 | 168 |
| Test S. clear | 100 | 80 | 80 |
| Cloudy | 405 | 284 | 248 |
| Train. Set cloudy | 177 | 174 | 201 |
| Test S. cloudy | 228 | 110 | 47 |

clear and cloudy spectra for each case study is provided. In the table also the number of clear and cloudy simulations used for defining the training sets and for the test sets is reported.

The capability to extract information content from the FIR is evaluated by applying the same procedure as before (see Sections 4.1 and 4.2).

5     Classification results obtained using 60 different training sets are shown in Fig. 13 for the 3 considered cases (MIDWIN, POLWIN and POLSUM respectively from the upper to the lower panel). Only results using the distributional approach are presented since CIC, on average, produces higher scores when run in this configuration.



It is shown that the classification scores increase by adding FIR channels to the MIR ones: average DP values are reported in the plots. However, the correlation between CoI and DP is less significant for the present cases with respect to what was found for the simulations in the Tropics. Also the DPs obtained for the MIDWIN, POLWIN, and POLSUM cases are lower than the ones obtained for the Tropical case. This result might be caused by the mean larger temperature differences between

high tropical cirrus clouds and the surface with respect to what found for Mid Latitude and Polar cases. It should be also considered that the Polar regions (especially in winter season) present surface features (ice and snow on the ground) that have radiative properties similar to ice clouds and thus making the clear/cloud identification extremely challenging in particular when analysing a test set containing a large number of thin cirrus clouds as the one under consideration.

The POLSUM case (lower panel in Fig. 13) show an evident correlation between the DP and CoI when the full spectrum is accounted for. Moreover, in this case, DP values are on average larger than 0.7 for CoI larger than 0.85. Therefore a RETS is chosen for this case to be used in testing the ability of CIC to correctly classify clouds with different optical depths. In Fig. 14 all the cloudy cases present in the POLSUM test set are classified using features in the MIR only or from the full FORUM spectrum. Results of the classification are plotted as a function of the CSID and of the cloud optical depth. It is noted that

the majority of the cloudy cases (also for optically thick clouds) are missed by CIC when relying on MIR channels only (left panel of Fig. 14). The scores improve significantly when the full spectrum is accounted for (right panel of the same figure). Nevertheless, optically thin clouds (mostly sub-visible cirrus clouds whose $OD < 0.03$) are still misclassified. The misclassified cases do not show any relation with the type of polar surfaces accounted for in the simulations (fine snow, medium snow, coarse snow or ice).

### 4.4 Computational time

In this section, a study of the computational time required to calculate the CoI is performed. The CoI computational time is a good indicator of the speed of the core algorithm, since its computation requires the SI computation of all training set elements and the additional optimisation needed for the distributional approach. Moreover, the CIC classification subroutine is the only

core routine whose duration depends on the number of features used for the classification. The current version of CIC algorithm is implemented in MATLAB programming language (https://www.mathworks.com/products/matlab.html).

When elementary approach is selected the CoI computation is not necessary. For this reason, the sensitivity study made in this section can also be interpreted as a study of the distributional approach computational cost and results represent an upper limit of the computational time for a classification process in any configuration.

In the upper panel of Fig. 15 results concerning the computational time of CoI as a function of the number of features ($N_{feat}$) and of the number of training set elements (here indicated with $T_{tot} = T_{clear} + T_{cloudy}$) are reported. Times are referred to computations performed on a machine with processor intel i5 (4 cores) and 4Gb RAM. It is shown that the computational time increase non-linearly with $N_{feat}$.




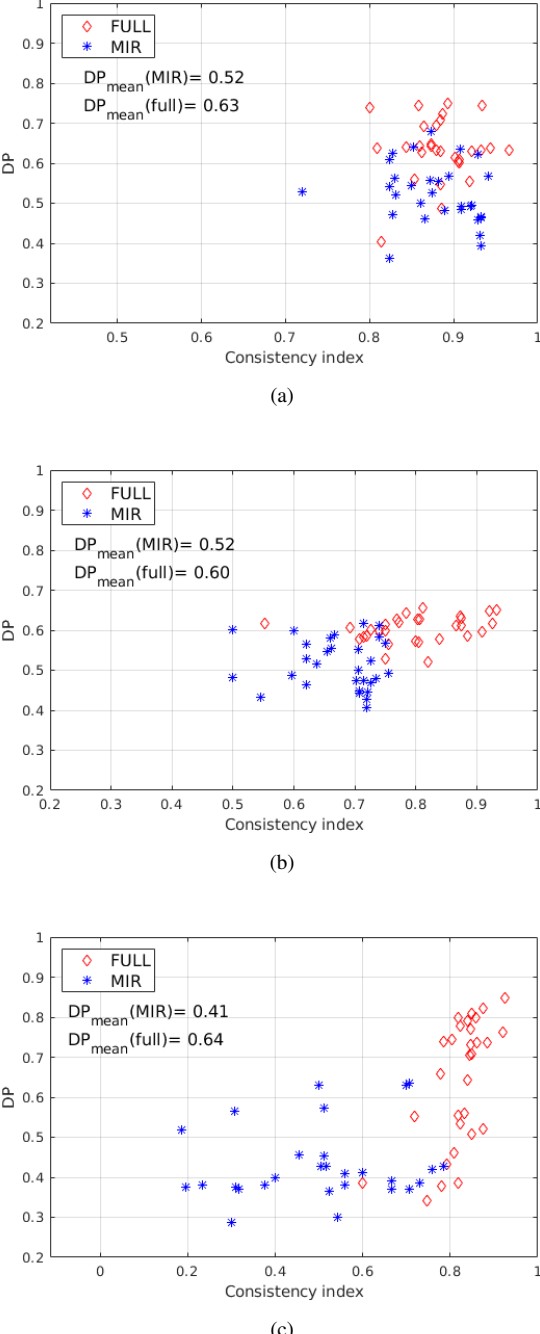

(a)

(b)

(c)

**Figure 13.** Clear/cloud classification DP using the distributional approach for the MIDWIN case (upper panel), the POLWIN case (middle panel) and POLSUM (lower panel). Data are plotted as function of CoI and 60 TraNC are used for each case.



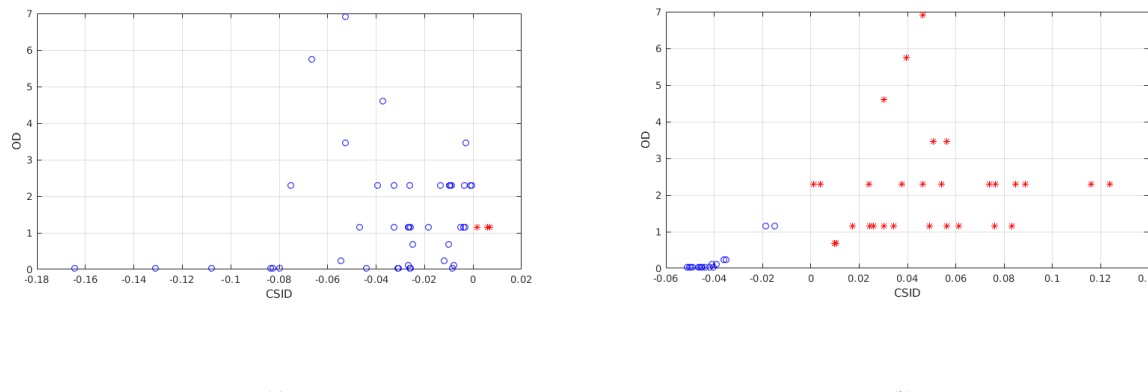

(a)                                                        (b)

**Figure 14.** Correctly classified (red asterisks) and misclassified (blue circles) cloudy sky spectra from the Polar Summer (POLSUM) test set. Data are plotted as a function of the corrected similarity index difference (CSID) and cloud optical depth (OD) and the distributional approach is considered. In the left panel only the MIR features are accounted for while on the right panel the full spectrum is exploited in the classification.

The algorithms for solving linear algebra problems cannot have a linear time complexity (computational complexity describing the time to run the algorithm). A lower bound is given by Raz (2003), where it is found a time complexity lower bound of $\mathcal{O}(n^2 log(n))$ for square matrix multiplication. It is demonstrated (Demmel et al. (2007)) that the same time complexity bound applies to most other linear algebra problems, including eigenvector computation, as performed by CIC.

CoI computation consists of three main subroutines: factor indicator function computation and its minimisation (see Eq. 8), the SI computation (see Eq. 7), and $shift_{opt}$ determination (Eq. 18). Among the three, the computational cost of the SI is the highest, since the other two only perform a limited and fixed number of cycles involving simple arithmetic operations. This subroutine computes the similarity indices for all $T_{tot}$ training set elements. SI computation requires the costly calculation of

10  the TREM matrices (see Eq. 3). TREM matrices need covariance computation, whose time complexity is $\Omega(N_{feat}T_{tot}^2)$, and eigenvector computation, whose time complexity is $\Omega(T_{tot}^3)$. Therefore, SI computation has an approximate time complexity of $\Omega(N_{feat}^2T_{tot}^2 + N_{feat}T_{tot}^3)$. The results of the computation of the SI parameter as a function of $T_{tot}$ and for a multiple number of $N_{feat}$ are reported in the lower panel of Fig. 15.

15  The running time of the remaining two routines (the one that computes the indicator function minimisation and the one that finds the $shift_{opt}$ parameter) is very limited and less than 0.3 seconds for any of the configurations accounted for $N_{feat}$ and $T_{tot}$.





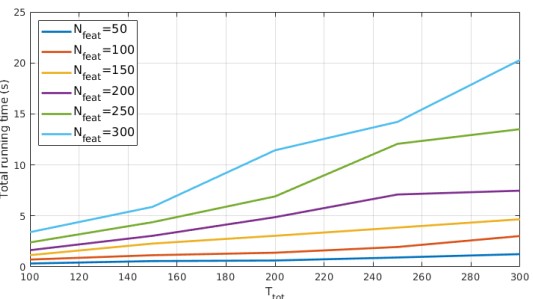

(a)

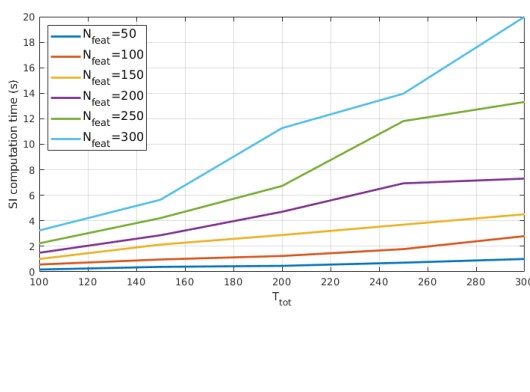

(b)

**Figure 15.** CoI coputational time (upper panel pane) and SI computational time (lower panel) as functions of training set size ($T_{tot}$) and of the number of features used ($N_{feat}$). A not linear relationship between the variables is observed.

Note that the Fourier Transform Spectrometer of the FORUM mission will perform one measurement every approximately 12 seconds (as stated in the FORUM Proposal for Earth Explorer Mission EE-9 'Fast Track' Earth Observation Envelope Programme June 2017) and thus the CIC algorithm could be run operatively on mission data.

# 5   Conclusions

5   A new methodology for spectra classification (named CIC: Cloud Identification and Classification) is presented. CIC is a very fast machine learning algorithm based on principle component analysis that depends on a limited number of user defined free parameters.



The algorithm exploits a training set composed of two groups of spectra (each group is a class). The training set elements should represent the observed variability within the classes and thus should include a sufficient number of spectra capable to characterise the radiative features encountered in the area of study. At the same time, they should be 'sensible' to the addition of new elements with spectral characteristics that are not present in the training set groups. Typically a total number of 100

spectra is sufficient to well characterise the clear sky and cloudy sky training set groups for each latitudinal belt (Tropics, Mid Latitudes, poles) and season. Mathematically the algorithm defines the similarity of each test set spectrum with each class and thus provide a classification. CIC, firstly, computes the eigenvectors of the covariance matrix of each training set class. Secondly a test set element is analysed by adding it to each training group (the clear and cloudy sky). The new eigenvectors of the extended covariance matrix (formed by the training set elements of one class plus the test set element) are computed. An

index of similarity is derived for the test set spectrum with respect to the two groups of the training set. It is assumed that a clear sky spectrum, added to the set of spectra defining the clear sky training set, do not significantly modify the group's principal components while some significant modifications will be detected if the clear sky spectrum is added to the cloudy sky training set. Viceversa is for a cloudy sky spectrum of the test set. Similarity indexes are thus defined to quantify the modification of the principal components of the training set when a new element (of the test set) is added. Based on these indexes the element

is associated to one of the two considered classes.

CIC can be run by adopting two approaches: the *elementary* and the *distributional* one. The first one is more intuitive and straightforward: the classification for each test set spectrum is done by comparing the similarity indexes computed with the two classes of the training set. The second one requires an additional a-priori optimisation process, at very low computational extra cost (Subsection 4.4). The optimisation is based on the definition of the Consistency Index (CoI) that is somehow inter-

pretable as the detection performance of the algorithm applied to the training set itself. Therefore, if the training set represents natural variability sufficiently well, the CoI serves as a performance forecaster. When optimisation is applied higher scores are obtained, as measured by the increased detection performance (DP, see Section 4) parameter.

The CIC is tested against a large synthetic dataset computed to simulate high spectral resolution radiance from satellite, specifically as possibly observed by the Earth Explorer Fast Track 9 candidate mission FORUM (Far Infrared Outgoing Radia-

tion Understanding and Monitoring). The measured FORUM radiance covers the 100-1600 $cm^{-1}$ spectral band (thus including the under-explored far infrared part of the spectrum) with a nominal spectral resolution of 0.3 $cm^{-1}$ and a goal noise of 0.4 $mW/(m^2 sr cm^{-1})$ in the 200-800 $cm^{-1}$ interval and 1 $mW/(m^2 sr cm^{-1})$ outside. The simulations are performed by using multiple surface properties, atmospherics profiles and different cloud features for liquid, mixed phase and ice clouds (including multiple ice habits). Simulations show that the far infrared part of the spectrum is particularly sensible to many atmospheric

parameters, such as upper tropospheric temperature and water vapour and to cloud geometrical and microphysical properties.

The dataset is divided in subsets in accordance with the latitudinal belt (and season for the Mid Latitudes) and the CIC is applied by accounting for different configurations. Results show that the CoI can be used to optimise the training set and that, statistically, the distributional approach is better performing than the elementary one. The code is also used to point out the additional information content derived from the analysis of the far infrared part of the spectrum with respect to the mid infrared

only. At this regards it is shown that the overall detection performances increase when the radiance spectra in the far infrared





are accounted for. In particular, the radiance exiting the 238-545 $cm^{-1}$ is used to improve the cirrus detection performances in almost all the atmospheric conditions (latitudinal belt and season). Very thin cirrus clouds (i.e. sub visible cirrus) are better detected when exploiting the full FORUM spectrum than the mid infrared part of the spectrum only. It is shown that, in tropical regions, the detection performances exploiting the full spectrum can be very high (higher than 0.9 for the present dataset that

is very challenging for the large presence of sub visible cirri) when the appropriate training set is selected. It is finally noted as clear/cloud spectra identification performances decrease when moving from Tropics to Poles mostly due to the decreased sensitivity of cloudy spectra because of the colder atmospheric and surface temperatures and the increased similarities in the surface and cloud radiative properties.

In the present work, CIC functionalities are illustrated for cloud detection application in presence of high spectral resolution

far and mid infrared radiance observations. Nevertheless, the same algorithm can, in principle, be implemented to work with different kind of data (i.e low spectral resolution data) and also to perform sub-classifications, such as cloud phase identification.

*Code and data availability.* The CIC source code is available on request from Sbrolli Iacopo who is the software developer of the algorithm.

*Competing interests.* The authors declare that they have no conflict of interest.

*Acknowledgements.* The present work is in preparation of the FORUM mission. FORUM related studies are supported by projects of the Italian Space Agency (ASI) and the European Space Agency (ESA).




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
