# Peer review of "Cloud identification and classification from high spectral resolution data in the far and mid infrared"

_Atmospheric Measurement Techniques, 2019_

## Referee Comment (RC1) · Anonymous Referee #1 · 13 Mar 2019

**General Comments**

The manuscript introduces a new algorithm for cloud identification and classification that is used on a simulated dataset for the upcoming ESA mission FORUM. The authors present their intentions and then proceed to explain how the experiment is performed, finally showing results and performance analysis, an important aspect for an algorithm that aims at being operational. The authors also show a good case study, albeit only simulated, for the importance of using the far infrared in conjunction with the usual 4-15 microns interval that is found in most hyperspectral sensors. The topic is certainly of interest and fits within the scope of the journal. Overall, the work is well organized. I recommend it for publication once the comments below have been addressed.

[Figure]

**Major Comments**

The paper makes an extensive use of acronyms, most of them defined within the manuscript and not always intuitive. This forces the reader to keep searching the text for their meaning, which distract from the content of the manuscript. If possible I would recommend to trim down the number of acronyms and use them only when necessary. In any case, adding an appendix with a list of acronyms can help the reader, who can at least find them in one place instead of scrolling continuously throughout the text.

Section 4

I'm a bit surprised by the findings here for the tropics. I would have expected the TraNCs configuration to be the opposite (i.e. more cloudy than clear spectra) since I'd imagine more cloudy components would help better capture the higher variability of the cloudy spectra. I'd like to hear if the authors have any thoughts on this. Out of curiosity, does this still hold true for mid latitudes and polar regions? in 4.3 the authors say that they generated the midlatitude and polar TraNCs with the same methodology used for the tropics, but it's not clear to me whether you used the same "70 clear/30 cloudy" split. If I understand correctly Table 6, this is not the case and you should at least mention that the tropical case and the non tropical ones are different in their training dataset (cloud/clear ratio).

Section 4.2

I wonder if excluding (some of) the FIR channels close to the $CO_2$ absorption would improve the DP even further. Do the authors have any comments or have they performed any tests with this case?

**Minor Comments**

Here below a list of typos and other minor corrections:
p1, l20: Please change "sensible" with "sensitive. There are a few other instances of this, listed below. Please, search the text, in case I missed some.

p2, l7: it is shown that **by** using. . .

p2, l10: REFIR-PAD, please define the acronym

p3, l3: this is a forward reference, the text redirect to a section down the line, please define the acronym SID here.

p3, l19: built

p3, 32: inconsistent capitalization

p5, l2: cloud, singular, the plural is determined by the adjective

p5, l2: either change "input" to "inputs" or "are" to "is"

p5, l7: **is** reported

p11, l24: the reference "Turner et al 2005" is incorrect, it should be Turner et al. 2006. Please change it

p16, l5: **In** this regard

p17, l15: either "performance" is plural or the verb "to be" singular

p17, 19: to **perform**

p17, l26: please use period instead of comma as decimal separator

p19, l16: . . . **are** provided for . . .

p19, l24: remove one of the articles: "the the"

p19, l30: depend**s**

p21, table 4: am I missing something or the RETS number don't add up? the total to me seems 25, not 30. Please explain.

p22, l1: . . . the number of features used in the FIR is reported

p25, l27: When **the** elementary. . .

p25, l33: G**B** the letter B should be capitalized (b: bits; B: bytes)

p25, l34: increase**s**

p27, l2-3: please rearrange the sentence to follow the subject + verb structure (i.e.: where a time complexity lower bound of O(. . .) is found for square matrix multiplication)

p29, l3: change "sensible" with "sensitive:

p29, l7: provide**s**

p29, l11: . . . the clear sky training set, do**es** not significantly modify . . .

p29, l25: spectral band**s**
p29, l29: again, change "sensible" with "sensitive"
p29, l35: **In** this regard
* * *

---

## Author Comment (AC1) · 12 Apr 2019

Reply to Anonymous Referee #1 concerning the AMT discussion paper entitled "Cloud identification and classification from high spectral resolution data in the far and mid infrared" by Tiziano Maestri et al.
In yellow our point by point responses to the Referee.
We would like to thank the Referee for the positive comments and above all for the suggestions and requests of clarifications that make the article much clearer and more readable.

**Anonymous Referee #1**

**General Comments**
**The manuscript introduces a new algorithm for cloud identification and classification that is used on a simulated dataset for the upcoming ESA mission FORUM. The authors present their intentions and then proceed to explain how the experiment is performed, finally showing results and performance analysis, an important aspect for an algorithm that aims at being operational. The authors also show a good case study, albeit only simulated, for the importance of using the far infrared in conjunction with the usual 4-15 microns interval that is found in most hyperspectral sensors. The topic is certainly of interest and fits within the scope of the journal. Overall, the work is well organized. I recommend it for publication once the comments below have been addressed.**

**Major Comments**
**The paper makes an extensive use of acronyms, most of them defined within the manuscript and not always intuitive. This forces the reader to keep searching the text for their meaning, which distract from the content of the manuscript. If possible I would recommend to trim down the number of acronyms and use them only when necessary. In any case, adding an appendix with a list of acronyms can help the reader, who can at least find them in one place instead of scrolling continuously throughout the text.**

True. A list of acronyms is inserted at the beginning of the article. Some acronyms are also eliminated since they were appearing just once along the article.

List of acronyms

BT - Brightness Temperature
CoI - Consistency Index
CIC - Cloud Identification and Classification
CSID - Corrected Similarity Index Difference
DP - Detection Performance
ETR - Extended Training Set
ETREM - Extended Training Set Eigenvector Matrix
FIR - Far InfraRed
FN - False Negative
FP - False Positive
FORUM - Far-infrared Outgoing Radiation Understanding and Monitoring
IBEC - Information-BEaring principal Component
IG2 - Initial Guess database number two

ILS - Instrumental Line Shape
IND - INDicator function
LbLRTM - Line by Line Radiative Transfer Model
MIDWIN - MID-latitude WINter
MIR - Mid InfraRed
MT_CKD - Mlawer, Tobin, Clough, Kneizys and Davies water vapor continuum model
OD - Optical Depth
PCA - Principal Component Analysis
POLSUM - POLar SUMmer
POLWIN - POLar WINter
POSCO - A POsteriori classification SCOre
PRISCO - A PRIori classification SCOre
RE - Real Error
REFIR-PAD - Radiation Explorer in the Far InfraRed: Prototype for Applications and Development
RETS - REference Training Set
RFTS - Refir Fourier Transform Spectrometer
TN - True Negative
TP - True Positive
TR - Training Set
TraNC - Training set Number Configuration
TREM - Training Set Eigenvector Matrix
TROSUM - TROpical SUMmer
TROWIN - TROpical WINter
SI - Similarity Index
SID - Similarity Index Difference

**Section 4**

**I'm a bit surprised by the findings here for the tropics. I would have expected the TraNCs configuration to be the opposite (i.e. more cloudy than clear spectra) since I'd imagine more cloudy components would help better capture the higher variability of the cloudy spectra. I'd like to hear if the authors have any thoughts on this.**

We agree with the Reviewer about the fact that this result is counterintuitive. The result is mostly related with the phase space volume occupied by both the clear and cloud training sets that have to be accurately defined in order to provide a balanced SI. In general, clear sky spectra occupy a smaller phase space volume with respect to the cloudy sky set of spectra. Thus, again in general, a larger number of elements should be included in the clear sky training set to make it less sensitive to the addition of an extra element from the test set. In this way the sensitivities of the clear sky training set and cloudy sky training set are of the same order and the final SI is well balanced. This problem happens when the training sets elements are chosen randomly from the whole dataset (or a subset of the whole dataset) instead of being accurately selected in order to fully cover the natural variability of the geographical area and season accounted for. During our tests (shown in the article) we preferred to adopt the random selection approach. The strategy of randomly select the training set spectra and the use of a statistical approach (multiple clear/cloud identifications are performed) avoid to obtain results that are linked to a specific training set choice.

Note that also the 50% cloud- 50% clear sky spectra configuration for the training set provides very high scores in many cases (cyan dots in figure 10 for example). In fact, in further investigations, we have tested that for accurately prepared training sets (elements could be manually selected and with extreme cases also included) the optimal number of the clear and cloudy elements is comparable. Note also that, in the current study, we only account for three TraNCs. Maybe there are different compositions that statistically works better than those proposed. Nevertheless, it is not one of the goals of the paper to find the number composition of training set (TraNCs) that provides the best performance when the elements are chosen randomly. The goal of the process is simply to have a requirement that we can apply in the application of CIC to the full test set.

A sentence has been added to the text at the beginning of Section 4.1 "The optimisation applies when the training set elements are randomly chosen from a large subset of the whole dataset. Results could be different if training set elements are manually selected in order to cover the natural variability of the cloud and clear sky spectra encountered for each latitudinal belt and season."

**Out of curiosity, does this still hold true for mid latitudes and polar regions?**

The same results, (not shown) have been found for the midlatitude and polar cases: the DP is statistically higher if $T_{cle} > T_{clo}$. This holds (again) when the training set elements are randomly chosen from dataset.
A sentence has been added to highlight the generality of the result (beginning of section 4.1).

**in 4.3 the authors say that they generated the midlatitude and polar TraNCs with the same methodology used for the tropics, but it's not clear to me whether you used the same "70 clear/30 cloudy" split. If I understand correctly Table 6, this is not the case and you should at least mention that the tropical case and the non tropical ones are different in their training dataset (cloud/clear ratio).**

Yes, the TraNCs for Polar and ML are chosen with the same composition (70-30). This has now been better clarified in the article with a new sentence which also highlights that the spectra are randomly selected from the dataset (of course with reference to latitude belt and season).
Now in the article (section 4.3):
"Mid Latitude winter (MIDWIN), Polar winter (POLWIN) and Polar summer (POLSUM) cases are presented. The TraNCs used are generated with the same methodology outlined for the Tropical case thus they are composed of 70 clear sky spectra and 30 cloudy sky spectra randomly chosen from a large subset of the full dataset. Spectra used to define the training sets are not inserted in the test set.
In Table 6 a summary of the total number of clear and cloudy spectra for each case study is provided. In the table the number of clear and cloudy simulations used for defining the training sets and for the test sets is also reported."

This should clarify the meaning of the Table 6. Nevertheless, the Table 6 caption is also changed for the same purpose: "MIDWIN, POLWIN, and POLSUM data set. For clear and cloudy condition

the table columns report the total number of simulations (spectra), the number of spectra used to define the training sets and the number of cases used as test sets."

**Section 4.2**
**I wonder if excluding (some of) the FIR channels close to the CO2 absorption would im- prove the DP even further. Do the authors have any comments or have they performed any tests with this case?**

Yes, probably the Reviewer is right but we haven't tested that. We decided to focus on the FIR part of the spectrum and to highlight the additional information content that is provided by FIR without analysing in details all the rest of the MIR.
It has to be noted that our approach is statistical: we performed a large number of runs for multiple training sets. The strategy was followed in order to avoid to obtain results that are specifically dependent on the training set itself. The same reason lays behind the choice of selecting the training set randomly for a large subset of the full dataset. This strategy implies that each analysis requires a large number of runs and thus it takes a large amount of times to arrange the experiment.
Thus, in summary, even if we agree with the Reviewer that it would be very interesting to test the effect of all the co2 channels on the DP, we think that it would not fit the goals of the present research that are:
- to describe a new methodology for cloud identification and classification and
- to highlight the key information derived from the fir

**Minor Comments**
**Here below a list of typos and other minor corrections:**
**p1, l20: Please change "sensible" with "sensitive. There are a few other instances of this, listed below. Please, search the text, in case I missed some.**
done
**p2, l7: it is shown that by using. . .**
done
**p2, l10: REFIR-PAD, please define the acronym**
done
**p3, l3: this is a forward reference, the text redirect to a section down the line, please define the acronym SID here.**
We got rid of reference the SID here and simply refers to a univariate distribution.
**p3, l19: built**
done
**p3, 32: inconsistent capitalization**
corrected to ERA-Interim reanalysis
**p5, l2: cloud, singular, the plural is determined by the adjective**
corrected
**p5, l2: either change "input" to "inputs" or "are" to "is"**
corrected
**p5, l7: is reported**
done
**p11, l24: the reference "Turner et al 2005" is incorrect, it should be Turner et al. 2006. Please change it**

done

**p16, l5: In this regard**

done

**p17, l15: either "performance" is plural or the verb "to be" singular**

done

**p17, 19: to perform**

corrected

**p17, l26: please use period instead of comma as decimal separator**

done

**p19, l16: ... are provided for ...**

changed

**p19, l24: remove one of the articles: "the the"**

done

**p19, l30: depends**

corrected

**p21, table 4: am I missing something or the RETS number don't add up? the total to me seems 25, not 30. Please explain.**

Right. We had erroneously inserted an old Table. The correct values are now inserted.

**p22, l1: . . . the number of features used in the FIR is reported**

corrected

**p25, l27: When the elementary. . .**

done

**p25, l33: GB the letter B should be capitalized (b: bits; B: bytes)**

corrected

**p25, l34: increases**

corrected

**p27, l2-3: please rearrange the sentence to follow the subject + verb structure (i.e.: where a time complexity lower bound of O(. . .) is found for square matrix multiplication)**

corrected

**p29, l3: change "sensible" with "sensitive:**

done

**p29, l7: provides**

corrected

**p29, l11: . . . the clear sky training set, does not significantly modify . . .**

corrected

**p29, l25: spectral bands**

done

**p29, l29: again, change "sensible" with "sensitive"**

done

**p29, l35: In this regard**

done

[revised manuscript text omitted]

---

## Referee Comment (RC2) · Anonymous Referee #2 · 30 Apr 2019

General Comments

The authors present the new cloud identification and classification algorithm CIC for far- and mid-infrared radiance measurements. The method is specifically designed for analysis of the ESA Earth Explorer candidate mission FORUM. Overall, I found this to be a scientifically sound study, which should be of interest for readers of AMT. The manuscript could be a bit more concise and some English language editing may be needed, but it is mostly clear. I would recommend the paper for publication in AMT.

Specific Comments

page 1, line 2-3: Saying the method uses 'a univariate distribution and a threshold' without additional information may not be clear to the reader. A distribution of which

variable or a threshold of which quantity?

page 1, line 1-10: I would suggest adding 1-2 sentences and numbers providing information on the accuracy and performance of the new CIC algorithm to the abstract.

page 3, line 3: An univariate distribution of which quantity?

page 3, line 29-32: It may not be too important to mention here that ERA-Interim data can be retrieved via a web interface. Saying 'surface height' is computed from temperature, pressure, and geopotential heights is a bit confusing, as surface height (or surface geopotential) is already a parameter in the ERA-Interim database. Do you mean 'geometric height' or 'height above the surface'?

page 4, Fig. 1: The RTX acronym/code was not introduced at this point. Also, the reader will not know what 'OD_deflt' means.

page 5, line 14-19: Is noise to be expected the leading error of the FORUM measurements? Is there a reference (e.g. an ESA report) available, providing more detailed information on the FORUM instrument? Instead of 'ideal measurement noise' perhaps say 'nominal measurement noise'?

page 6, Table 2: Can you provide a rough estimate on the percentage error of the noise estimates? The percentage errors might be rather large for the FIR part of the spectrum because measured radiances are quite low?

page 6, Table 3: 'Polar regions' covers both, polar winter and polar summer. Perhaps this should be split into the corresponding cases?

page 11, line 17-19: This is providing a specific definition of the similarity index used in this study. Did you consider any other potential measures of similarity? Was there a specific rationale to chose this definition?

page 13, Fig. 7: Make plots the same size.

page 15, line 9: Perhaps say that 'cardinality' means 'number of elements of the set',

which is simpler for non-mathematicians.

page 16, line 24 to p17, line 12: If the POSCO is not needed or evaluated during the rest of the manuscript, it may not have to be introduced here at all?

page 17, line 23: If I understood correctly, only a subset of all FORUM channels is considered. Is there any exploitable information content in the unused parts of the FORUM spectrum?

page 18, line 1: At this point, it may not be clear why a correlation between the CoI and DP can be found? From Fig. 9 this correlation does not become very evident, as these plots basically seem to show point clouds with some outliers? What is the correlation coefficient between CoI and DP?

page 19, line 9-14: What is the rationale for creating different TraNCs? Is this meant to reflect a priori knowledge on real cloud distributions in the atmosphere?

page 19, line 1: Was the tropical test case selected for presentation because it is the most difficult or most simple case?

page 19, line 1-2: I was wondering if it is sufficient to use only one test set (per class) for validation of the classification methods. The classification method might be tuned to work best only for the specific test set and may show different results for another test set. Did you consider to rerun the analysis with a different choice of test sets?

page 20, Fig. 10: Do the results presented here change if another test set is used?

page 21, Table 4: Perhaps choose more even bins for OD, i.e., 0.1 to 0.3 and 0.3 to 1 rather than 0.1 to 0.5 and 0.5 to 1.0?

page 21, line 13: Which $CO_2$ band is meant?

page 22, line 10-12: Can you explain the initial decrease in performance when the first FIR channels are added to the classification?

page 23, Fig. 11: Perhaps it could help to add a few more data points to this figure, to better understand what is happening when the first FIR channels are added to the classification?

page 23, line 12-13: I am afraid I do not understand the sentence 'Note that the DP value is the minimum...' in this context. Can you explain it a bit better?

page 24, Fig. 12: I have some difficulty identifying any clear correlation between OD and CSID from these figures.

page 27, line 1-4: This sounds as if almost every linear algebra problem can be solved with an O(n^2 log n) algorithm, which is too general. I would rephrase this a bit and just refer to the algorithms used in this study.

page 29, line 21-22: Can you quantify this? How much higher are the scores?

page 30, line 2-3: Can you quantify this? How much better was the detection of thin cirrus?

All Figures: Please check and enlarge the font size of the labels to make them better readable.

Technical Corrections

page 1, line 8: change 'i.e' to 'i.e.'

page 2, line 10: introduce REFIR-PAD acronym

page 2, line 19-20: check that acronyms are properly introduced

page 2, line 31: change 'mostly widely' to 'most widely'

page 2, line 32: change 'Feature' to 'feature'

page 2, line 33: change 'Spectral Fitting' to 'spectral fitting'

page 3, line 12: change 'profiles' to 'conditions'

page 3, line 14: rephrase to 'the CIC algorithm'

page 3, line 32: change 'Era-Interim' to 'ERA-Interim'

page 4, line 8: change 'fulfil this information' to 'add information'

page 4, line 11: change 'of the spectrum representatives of ' to 'representative of'

page 4, line 15: change 'Scattnlay' to 'ScattNLay'

page 4, Table 1: change 'Cloud properties' to 'Cloud property'

page 6, line 12: change 'a presence' to 'the presence'

page 7, line 3: change '(CIC' to '(CIC)' (or delete)

page 11, line 4: change 'line' to either 'row' or 'column' (as applicable)

page 11, line 19 and page 12, line 15: remove extra brackets () for ETREM term

page 16, line 2: perhaps replace 'an algorithm' by 'a cloud classification algorithm' to be more specific?

page 16, line 19: change 'i,:' to 'i:'

page 17, line 15: change 'are evaluated' to 'is evaluated'

page 17, line 24: change 'but a small' to 'except for a small'

page 19, line 19: fix 'The the'

page 23, line 1: rephrase to 'a function'

page 23, line 8: rephrase to 'The left plot'

page 25, line 11: DP values are ... than 0.7 *and* for CoI

page 25, line 27: rephrase to 'When the elementary'

page 25, line 33: change 'intel' to 'Intel'

page 27, line 2: delete 'it is found'

page 27, line 12: change 'a multiple number' to 'different numbers'

page 28, Fig. 15: change 'not linear' to 'non-linear'

page 28, line 1: change 'every approximately' to 'about every'

page 28, line 5: rephrase to 'cloud spectra detection and classification'

page 29, line 6: change 'defines' to 'evaluates'

page 29, line 7: change 'provide' to 'provides'

page 29, line 11: change 'do' to 'does'

page 29, line 19: change 'somehow interpretable as' to 'related to'

page 29, line 23: rephrase 'computed to simulate'

page 29, line 33: change 'point out' to 'assess'

page 30, line 1: rephrase to '238-545 cmˆ-1 wavenumber range is improving the'

page 30, line 5-6: change 'noted as' to 'noted that'

---

## Editor Comment (EC1) · Lars Hoffmann (Editor) · 13 May 2019

Dear authors,

for the sake of transparency, I've been asked to post three additional comments on your paper provided by a colleague as an editorial comment.

Please consider these comments in the revision of your manuscript:

Page 1 Line 23. The authors probably intended to cite Serio et al 2008 (doi:10.1364/OE.16.015816) and not Serio et al. 2000

Page 3 Line 24. Probably the version of LbLRTM used is v12.7 and not v2.7

Page 7 Line 10. Parenthesis ) is missing.

[Figure]

Thank you and best regards

Lars Hoffmann

---

## Author Comment (AC3) · 27 May 2019

Reply to Anonymous Referee #2 concerning the AMT discussion paper entitled "Cloud identification and classification from high spectral resolution data in the far and mid infrared" by Tiziano Maestri et al.
In yellow our point by point responses to the Referee.
We would also like to thank the Referee for the positive comments and above all for the suggestions and requests of clarifications that helped in making the article much clearer and more readable.

**Anonymous Referee #2**

**General Comments**
**The authors present the new cloud identification and classification algorithm CIC for far- and mid-infrared radiance measurements. The method is specifically designed for analysis of the ESA Earth Explorer candidate mission FORUM. Overall, I found this to be a scientifically sound study, which should be of interest for readers of AMT.**

We thank the Referee for the positive comment. The algorithm has been applied to high spectral resolution synthetic data simulating the FORUM mission but the methodology could be easily adapted to every satellite or ground sensor covering a large enough spectral band sensible to surface and atmospheric (clear and cloudy) features variations.
Currently, the CIC is applied to TAFTS and ARIES Fourier Spectrometers airborne data in collaboration with MetOffice and Imperial College (UK) within the Phase-A study of the Earth Explorer 9 ESA Fast Track mission and to the data from the interferometer REFIR-PAD that is ground-based at Dome-C station in the Antarctic Plateau.
A new sentence is added at the end of the Conclusions section to highlight the potentialities and easy adaptability of the CIC algorithm to new sensors and viewing geometries.
The new text is reported here for your convenience:
"…In the present work, CIC functionalities are illustrated for cloud detection application in presence of high spectral resolution far and mid infrared radiance observations. Nevertheless, the same algorithm can, in principle, be implemented to work with different kind of data (i.e low spectral resolution data) and also to perform sub-classifications, such as cloud phase identification. The CIC algorithm is easily adaptable to different viewing geometries and diverse high spectral resolution sensors. Currently, it is being tested against interferometric data in the far and mid infrared part of the spectrum collected by the airborne Tropospheric Airborne Fourier Transform Spectrometer (TAFTS, Canas et al. (1997)) and the Airborne Research Interferometer Evaluation System (ARIES, Wilson et al. (1999)) during the 2015 CIRCCREX (Cirrus Coupled Cloud-Radiation Experiment) campaign (Pickering et al., 2015) and to ground based data collected by the REFIR-PAD interferometer since 2012 from the Dome-Concordia station on the Antarctic Plateau (http://refir.fi.ino.it/refir-pad-domeC). "

**The manuscript could be a bit more concise and some English language editing may be needed, but it is mostly clear. I would recommend the paper for publication in AMT.**

The manuscript has been greatly improved also by applying all the grammar and typo corrections suggested by both referees and the Editor.

**Specific Comments**

**page 1, line 2-3: Saying the method uses 'a univariate distribution and a threshold' without additional information may not be clear to the reader. A distribution of which variable or a threshold of which quantity?**

The sentence is modified as follows:

"… CIC is a machine-learning algorithm, based on Principal Component Analysis, able to perform a cloud detection and scene classification using a univariate distribution of a similarity index which defines the level of closeness between the analysed spectra and the elements of each training datasets."

**page 1, line 1-10: I would suggest adding 1-2 sentences and numbers providing infor- mation on the accuracy and performance of the new CIC algorithm to the abstract.**

The performances are strictly related to the training and test sets accounted for thus a general statement is difficult to make. Anyway, a new sentence is added at the end of the abstract:

"… In particular, it is shown that hit scores for clear and cloudy spectra increase from about 70% up to 90% when far-infrared channels are accounted for in the classification of the synthetic dataset for tropical regions."

**page 3, line 3: An univariate distribution of which quantity?**

True.

The sentence is changed to:

"CIC is a machine learning algorithm based on Principal Component Analysis, which performs an identification and classification using a single threshold applied to a univariate distribution of a newly defined parameter called similarity index (see Section 3.2) which determines the relatedness with a specific class (training set)."

**page 3, line 29-32: It may not be too important to mention here that ERA-Interim data can be retrieved via a web interface. Saying 'surface height' is computed from temperature, pressure, and geopotential heights is a bit confusing, as surface height (or surface geopotential) is already a parameter in the ERA-Interim database. Do you mean 'geometric height' or 'height above the surface'?**

The sentence is re-phrased:

"This database, that provides 4 sets of data per day, is used to retrieve profiles of temperature, pressure, specific humidity, ozone mixing ratio, and surface geopotential height from which the geometric surface and atmospheric level heights are computed."

**page 4, Fig. 1: The RTX acronym/code was not introduced at this point. Also, the reader will not know what 'OD_deflt' means.**

The Figure has been redone and an extended caption is added and here reported:

[Figure]

"Figure 1. Software architecture (schematic) of the simulation process used to build the synthetic dataset for FORUM-like observations. Blue box are codes, red ones are auxiliary datasets. The FORUM box includes the Fourier Transform Spectroscopy and the simulation of the FORUM noise. OD stands for gaseous optical depths computed using the Line-by-Line Radiative Transfer Model LbLRTM v12.7 (Clough et al. (2005)). The Radiative Transfer X (RTX) is described in Bozzo et al. (2010). "

**page 5, line 14-19: Is noise to be expected the leading error of the FORUM measure- ments? Is there a reference (e.g. an ESA report) available, providing more detailed information on the FORUM instrument? Instead of 'ideal measurement noise' perhaps say 'nominal measurement noise'?**

NESR is the leading error of the FTS sensor of the FORUM mission. The mission is undergoing Phase A industrial studies and the updated mission requirements will be revealed during the Earth Explorer 9 User Consultant Meeting to be held in Cambridge, UK, 16-17 July 2019.  The mission proposal with preliminary sensor requirements is available on request from the corresponding author Tiziano Maestri (tiziano.maestri@unibo.it).

The sentence is modified as follows:

"In a subsequent step, a nominal noise equivalent spectral radiance (NESR), as reported in the FORUM Proposal RCEE9/027 to ESA, is added to the simulated radiances in order to produce a realistic FORUM observations dataset.

…

The spectral dependence and amplitude of the noise are derived from the technical specification of the Fourier Transform Spectrometer instrument described in the FORUM proposal to ESA (available on request) and reported in Table 2."

**page 6, Table 2: Can you provide a rough estimate on the percentage error of the noise estimates? The percentage errors might be rather large for the FIR part of the spectrum because measured radiances are quite low?**

A new sentence is added:

"The NESR values reported in the Table corresponds to a typical percentage noise of about 1% in the 200-800 $cm^{-1}$ wavenumber interval. The exact value depends on the specific wavenumber and observational conditions accounted for. Below 200 $cm^{-1}$ and above 1400 $cm^{-1}$ the percentage noise can be higher than 15% due to the low radiance values."

**page 6, Table 3: 'Polar regions' covers both, polar winter and polar summer. Perhaps this should be split into the corresponding cases?**

Since the Table presents the dataset divided for latitudinal region and since the number of cases in polar regions is the lowest we would prefer to keep the Table like it is.

The values are anyway reported here for the Referee:

|              | Clear sky | Clouds [Liq / Mixed / Ice (svc)] |
|--------------|-----------|----------------------------------|
| Polar summer | 248       | 284 [00 / 24 / 260 (68)]         |
| Polar winter | 244       | 248 [00 / 24 / 224 (64)]         |

**page 11, line 17-19: This is providing a specific definition of the similarity index used in this study. Did you consider any other potential measures of similarity? Was there a specific rationale to chose this definition?**

We have defined this index since it accounts for all the spectral features that characterize the physical signal. As you can note, the same weight is associated to the loadings of the $p^{th}$ principal components considered in the sum which defines the SI. This makes the CIC very sensible to any spectral signature introduced in the spectrum and thus able to capture the information content comprised in the far infrared part of the spectrum. A sentence is added in this regard:

"Using the same weight ($1/2P_0$) for each term of the sum makes the SI very sensible to any spectral signature presents in the spectrum. "

**page 13, Fig. 7: Make plots the same size.**

Done

**page 15, line 9: Perhaps say that 'cardinality' means 'number of elements of the set', which is simpler for non-mathematicians.**

Agreed and substituted.

**page 16, line 24 to p17, line 12: If the POSCO is not needed or evaluated during the rest of the manuscript, it may not have to be introduced here at all?**

Since the Detection Performance is based on the a-priori probability only we provided an explicit relation between PRISCO and POSCO for those who are more used to the a-posteriori probability.

**page 17, line 23: If I understood correctly, only a subset of all FORUM channels is considered. Is there any exploitable information content in the unused parts of the FORUM spectrum?**

True we have used channels in the 100-1300 $cm^{-1}$ range.

In the first part of the article some forward simulation results have been plotted to show the sensitivity of the radiance spectrum to the surface, atmospheric and cloud parameters. Results show that the 300-1330 cm$^{-1}$ part of the spectrum is the most sensible to cloud features variation. This is also shown in Maestri et al. (2019) for downwelling radiances and by means of a linear discriminant analysis.

An additional reason is due to the low FORUM signal-to-noise-ratio for wavenumbers above 1300 cm$^{-1}$ because of the low values of the spectral radiance and of the large nominal NESR at those wavenumbers (as discussed above). This is also true for wavenumber below 300 cm$^{-1}$ but we have decided to retain this extreme part of the spectrum in the study due to the scarcity of the study concerning the FIR found in literature. Also, it is shown how the detection performances reduces when channels below 238 cm$^{-1}$ are used and a possible explanation is related to the almost null radiance sensitivity to cloud features in that band.

**page 18, line 1: At this point, it may not be clear why a correlation between the CoI and DP can be found? From Fig. 9 this correlation does not become very evident, as these plots basically seem to show point clouds with some outliers? What is the correlation coefficient between CoI and DP?**

The purpose of the figures is not to show the correlation functionality between the two variables. The goal is to highlight that high detection performance is mainly reached for large consistency indexes. Note that the right panel DPs are higher than those found in the left panel that correspond to smaller CoI. This relation is observed also in Figure 10.

**page 19, line 9-14: What is the rationale for creating different TraNCs? Is this meant to reflect a priori knowledge on real cloud distributions in the atmosphere?**

Multiple training sets are created (by randomly selecting the elements) in order to show the generality of some aspects of the classification process. The main goal of the paper, apart from describing a new methodology, is to show that the far infrared part of the spectrum contains useful information that can be exploited to greatly improve the scene classification in absence of shortwave scattered radiation (i.e. Figure 9 and 13). This is demonstrated statistically using a large set of different training sets composed by randomly selected elements with the only constrain on the number of each class elements. If this was done using a single training set, with manually selected spectra, it would have been susceptible of subjectivity. As far as the different Training sets number composition (TraNCs), these are used to investigate if an optimal fraction of clear to cloudy spectra in the training sets exists for the specific datasets to which the classification is applied. It is shown that statistically the 70clear-30cloudy number composition works better (i.e. Figure 10) but very high scores are also obtained with the 50-50 number compositions. For specific observing conditions (i.e. the analysis of data from the 2015 CIRCCREX campaign mentioned above) we have generated clear and cloudy training sets with the same number of elements that are specifically created to reproduce the natural variability characteristic of the experimental site.

**page 19, line 1: Was the tropical test case selected for presentation because it is the most difficult or most simple case?**

Simulations are performed for all the cases. Classifications results are also shown for all the latitudes and seasons. The Tropics are used as example for the definition of an optimised training set because of the large number of synthetic spectra available and of the large variability of the simulated conditions spanning from hot clear desert scenes to warm low level liquid phase marine strati to very thin High-Troposphere cirri and dense hurricane's clouds.

Classifications are though performed also at Mid and Polar latitudes.

**page 19, line 1-2: I was wondering if it is sufficient to use only one test set (per class) for validation of the classification methods. The classification method might be tuned to work best only for the specific test set and may show different results for another test set. Did you consider to rerun the analysis with a different choice of test sets?**

We have used this subdivision only for the definition of an optimisation strategy and for the training set number composition. The methodology is applied to different test set when different latitudes and seasons are accounted for. The code algorithm is tested against very different conditions that are intended to reproduce most of the natural variability encountered all over the globe.

We also have seen that for specific observational conditions (like the ones encountered in the 2015 CIRCCREX campaign or the ones concerning the ground based data collected by the REFIR-PAD interferometer at Dome-Concordia station on the Antarctic Plateau) the classification results are much higher than those presented here that refers to a very challenging test set filled with large number of sub-visible-cirrus clouds that are very difficult to detect.

**page 20, Fig. 10: Do the results presented here change if another test set is used?**

As discussed above the results are probably slightly dependent on the test set. The results obtained in this section are functional to the more general results shown in the following figures (i.e. Figure 11 and 12). They serve to define an optimised training set that is then adopted to find the main results of the paper. Different training sets can be used as reference without changing the main result that states that the use of FIR channel improves the classification scores.

**page 21, Table 4: Perhaps choose more even bins for OD, i.e., 0.1 to 0.3 and 0.3 to 1 rather than 0.1 to 0.5 and 0.5 to 1.0?**

The Table has been changed due to a type spotted by the other Referee. The new Table 4 is here reported for your convenience. We would like to keep the number as they are also since the numbers of elements in the RETS between 0.1 and 1.0 is just 4.

| Training and test set cloud ODs | | |
|---|---|---|
| Range | RETS | Test set |
| $OD \leq 0.1$ | 8 (6.3%) | 126 |
| $0.1 < OD \leq 0.5$ | 3 (2.3%) | 131 |
| $0.5 < OD \leq 1$ | 1 (5.3%) | 19 |
| $1 < OD \leq 3$ | 14 (4.5%) | 308 |
| $3 < OD \leq 10$ | 0 (0%) | 21 |
| $OD > 10$ | 4 (6.8%) | 59 |
| Total | 30 (4.5%) | 664 |

**page 21, line 13: Which CO2 band is meant?**

Sentence is rephrased to:

"Thus the full spectrum spans over the 100-1300 $cm^{-1}$ spectral range with the exception of a limited wavenumber interval in the $v_2$ $CO_2$ band centred at around 667 $cm^{-1}$."

**page 22, line 10-12: Can you explain the initial decrease in performance when the first FIR channels are added to the classification?**

A new sentence is added:

"Results show that performance gradually improves for increasing number of FIR channels. In particular, there is a slight decrease in performance after adding the 12 channels closest to the $CO_2$ $v_2$ band centre which are mostly insensible to cloud parameters due to strong absorption of the carbon dioxide. The decrease is gradually offset by improvements obtained when channels in the [238.8–545.4] cm$^{-1}$ wavenumber range are added. "

**page 23, Fig. 11: Perhaps it could help to add a few more data points to this figure, to better understand what is happening when the first FIR channels are added to the classification?**

We think that the mechanism is clear and we have discussed it in the point above. The selection of the FIR channels ($N_{feat}$) follows the formula described in equation 30. We decided to start our selection at 640 cm$^{-1}$ in order to include channels with strong $CO_2$ absorption. We did not know a-priori which channels could have brought a sensible improvement in the classification scores.The major interest is focussed on the full FIR band at this stage.

**page 23, line 12-13: I am afraid I do not understand the sentence 'Note that the DP value is the minimum...' in this context. Can you explain it a bit better?**

Correct. The sentence has been re-phrased:

"Note that the DP value is the minimum between the hit rate computed for cloudy sky cases and clear sky cases (see equation 25) and thus is indicative of the CIC ability to correctly classify either clear and cloudy spectra."

**page 24, Fig. 12: I have some difficulty identifying any clear correlation between OD and CSID from these figures.**

The Figure is not intended to show a relation between OD and CSID. The plot wants to show the ability to classify the scene for different ODs. CSID is the classifier (larger than 0 we have a cloudy sky classification). On the left it is shown, that when using the MIR only, a large number of cloudy sky cases are classified as clear sky (blue circles). On the right it is shown that the number of False Positives is strongly reduced when using the full spectrum (MIR+FIR) and even if a single clear sky is misclassified and identified as cloudy (green x) the overall performance is greatly improved by a correct classification of the majority of cloudy sky cases.

**page 27, line 1-4: This sounds as if almost every linear algebra problem can be solved with an O(n^2 log n) algorithm, which is too general. I would rephrase this a bit and just refer to the algorithms used in this study.**

We have re-phrased:

"Raz (2003), shows than a lower bound for time complexity of matrix multiplication is O(n2log(n)) and Demmel et al. (2007)) demonstrated that the same time complexity bound applies to most other linear algebra problems, including eigenvector computation, as performed by CIC."

**page 29, line 21-22: Can you quantify this? How much higher are the scores?**

The value is added to the sentence:

"When optimisation is applied higher scores are obtained, as measured by the increased detection performance (DP, see Section 4) parameter that can reach values as high as 0.95. "

**page 30, line 2-3: Can you quantify this? How much better was the detection of thin cirrus?**
Some number are provided:
"The hit scores for cirrus clouds with optical depth less than 0.06 moves from about 25% when using the mid infrared only to about 60% when exploiting also the FIR part of the spectrum. It is shown that, in tropical regions, the overall detection performances exploiting the full spectrum can be very high (higher than 0.9 for the present dataset that is very challenging for the large presence of sub visible cirri) when the appropriate training set is selected. "

**All Figures: Please check and enlarge the font size of the labels to make them better readable.**
Most of the Figures have been resized

**Technical Corrections**
**page 1, line 8: change 'i.e' to 'i.e.'**
done
**page 2, line 10: introduce REFIR-PAD acronym**
done
**page 2, line 19-20: check that acronyms are properly introduced**
all acronyms are re-checked. A list of acronyms is provided at the beginning of the article.
**page 2, line 31: change 'mostly widely' to 'most widely'**
corrected
**page 2, line 32: change 'Feature' to 'feature'**
corrected
**page 2, line 33: change 'Spectral Fitting' to 'spectral fitting'**
corrected
**page 3, line 12: change 'profiles' to 'conditions'**
changed
**page 3, line 14: rephrase to 'the CIC algorithm'**
done
**page 3, line 32: change 'Era-Interim' to 'ERA-Interim'**
done
**page 4, line 8: change 'fulfil this information' to 'add information'**
changed
**page 4, line 11: change 'of the spectrum representatives of ' to 'representative of'**
changed
**page 4, line 15: change 'Scattnlay' to 'ScattNLay'**
changed
**page 4, Table 1: change 'Cloud properties' to 'Cloud property'**
changed
**page 6, line 12: change 'a presence' to 'the presence'**
changed
**page 7, line 3: change '(CIC' to '(CIC)' (or delete)**
changed
**page 11, line 4: change 'line' to either 'row' or 'column' (as applicable)**
corrected to "row"
**page 11, line 19 and page 12, line 15: remove extra brackets () for ETREM term**
removed

**page 16, line 2: perhaps replace 'an algorithm' by 'a cloud classification algorithm' to be more specific?**

Suggestion accepted

**page 16, line 19: change 'i,:' to 'i:'**

done

**page 17, line 15: change 'are evaluated' to 'is evaluated'**

The word "performances" also changed to "performance" to be coherent with Referee suggestion: "In this section the performance of the CIC cloud detection algorithm is evaluated for multiple atmospheric conditions."

**page 17, line 24: change 'but a small' to 'except for a small'**

done

**page 19, line 19: fix 'The the'**

corrected

**page 23, line 1: rephrase to 'a function'**

done

**page 23, line 8: rephrase to 'The left plot'**

**done**

**page 25, line 11: DP values are ... than 0.7 *and* for CoI**

Not clear to us what we should change. We have re-phrase the sentence

"Moreover, in this case, DP values are on average larger than 0.7 for CoI larger than 0.85."

To

"Moreover, in this case, DP values are on average larger than 0.7 when CoI are larger than 0.85."

**page 25, line 27: rephrase to 'When the elementary'**

done

**page 25, line 33: change 'intel' to 'Intel'**

done

**page 27, line 2: delete 'it is found'**

the sentence has been rephrased in accordance with previous comment of the Referee

**page 27, line 12: change 'a multiple number' to 'different numbers'**

done

**page 28, Fig. 15: change 'not linear' to 'non-linear'**

done

**page 28, line 1: change 'every approximately' to 'about every'**

done

**page 28, line 5: rephrase to 'cloud spectra detection and classification'**

done

**page 29, line 6: change 'defines' to 'evaluates'**

done

**page 29, line 7: change 'provide' to 'provides'**

done

**page 29, line 11: change 'do' to 'does'**

done

**page 29, line 19: change 'somehow interpretable as' to 'related to'**

changed

**page 29, line 23: rephrase 'computed to simulate'**

Not clear to us what we should change

**page 29, line 33: change 'point out' to 'assess'**

done

**page 30, line 1: rephrase to '238-545 cm^-1 wavenumber range is improving the'**
done
**page 30, line 5-6: change 'noted as' to 'noted that'**
done

[revised manuscript text omitted]